# NEURAL EXPLORATORY LANDSCAPE ANALYSIS FOR META-BLACK-BOX-OPTIMIZATION

**Zeyuan Ma**[1]**, Jiacheng Chen**[1]**, Hongshu Guo**[1]**, Yue-Jiao Gong**[1,*]
[1]South China University of Technology
{scut.crazynicolas, jackchan9345, guohongshu369,
gongyuejiao}@gmail.com

## ABSTRACT

Recent research in Meta-Black-Box Optimization (MetaBBO) have shown that meta-trained neural networks can effectively guide the design of black-box optimizers, significantly reducing the need for expert tuning and delivering robust performance across complex problem distributions. Despite their success, a paradox remains: MetaBBO still rely on human-crafted Exploratory Landscape Analysis features to inform the meta-level agent about the low-level optimization progress. To address the gap, this paper proposes Neural Exploratory Landscape Analysis (NeurELA), a novel framework that dynamically profiles landscape features through a two-stage, attention-based neural network, executed in an entirely end-to-end fashion. NeurELA is pre-trained over a variety of MetaBBO algorithms using a multi-task neuroevolution strategy. Extensive experiments show that NeurELA achieves consistently superior performance when integrated into different and even unseen MetaBBO tasks and can be efficiently fine-tuned for further performance boost. This advancement marks a pivotal step in making MetaBBO algorithms more autonomous and broadly applicable. The source code of NeurELA can be accessed at `https://github.com/GMC-DRL/Neur-ELA`.

## 1 INTRODUCTION

Black-Box Optimization (BBO) is a discipline in mathematical optimization characterized by the absence of both the closed-form expression of the target objective function and its derivative during the optimization process. Traditional BBO methods, such as Genetic Algorithm (Holland, 1975), Evolutionary Strategy (Hansen & Ostermeier, 2001a), and Bayesian Optimization (Shahriari et al., 2015), have been extensively studied in the past half century. They are applied to address a broad range of black-box scenarios, including automated machine learning (Akiba et al., 2019), bioinformatics (Tsaban et al., 2022), prompt tuning in large language models (Guo et al., 2024b). However, these methods often require manual tuning with deep expertise to ensure the optimal performance on unseen tasks. Recent advances in Meta-Black-Box Optimization (MetaBBO) (Xue et al., 2022; Lange et al., 2023; Chen et al., 2024; Guo et al., 2024a; Lange et al., 2024) leverage meta-learning (Finn et al., 2017) to streamline the traditional labour-intensive tuning process across different problem distributions. This effort not only reduces the need for expert-level knowledge, but also enhances generalization towards unseen problems. As depicted on the left side of Figure 1, MetaBBO generally employs a bi-level optimization paradigm (Ma et al., 2023). In the meta level, a control policy (often implemented as a neural network) is maintained to dictate timely algorithm configuration for the low-level BBO optimizer according to the low-level optimization status. Once obtained the algorithm configuration, the low-level BBO optimizer optimizes the target BBO problem by that configuration and returns a feedback to the meta-level policy the performance gain feedback. The meta-objective of MetaBBO is to meta-learn an optimal control policy which could achieve maximum accumulated performance gain across a problem distribution.

One of the core challenges to ensure the generalization performance of MetaBBO algorithms is to design an information-rich profiling system for the low-level optimization status (highlighted in yellow box on the left side of the Figure 1). Firstly, the optimization status must include the

---

*Yue-Jiao Gong is the corresponding author.

identification features to differentiate various optimization problems. This design is crucial to enable the meta-level neural policy to be aware of *what is the target optimization problem*. Secondly, the optimization status should align with the low-level optimization dynamics, which helps to inform the meta-level neural policy about *where the low-level optimization progress has reached*. Existing

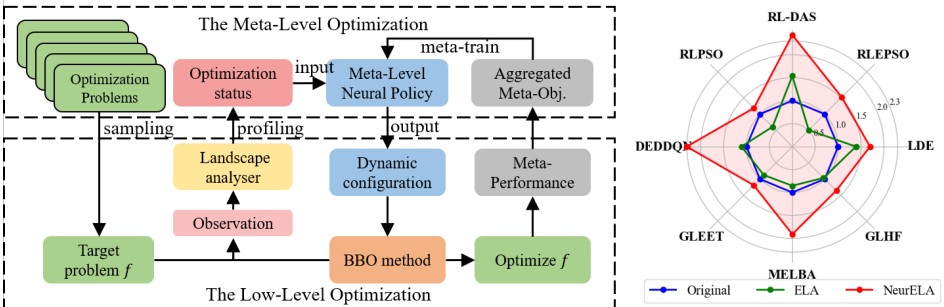

Figure 1: **Left**: The general workflow of MetaBBO algorithms, which follows a bi-level optimization paradigm. The landscape analyser timely profiles the low-level optimization progress, informing the meta-level neural policy to dynamically output desired configuration for the low-level BBO method. **Right**: The average optimization performance (larger is better) of integrating our pre-trained landscape analyser into diverse MetaBBO tasks (red line). The blue line denotes the original performance of the MetaBBO algorithms, while the green line denotes the performance of integrating the traditional exploratory landscape analysis features into the MetaBBO algorithms.

MetaBBO algorithms widely adopt Exploratory Landscape Analysis (ELA) (Wright et al., 1932; Mersmann et al., 2011) to profile the low-level optimization status. While the approaches provide a comprehensive understanding of problem and algorithm characteristics, they still face several limitations. 1) Many features derived from ELA are highly correlated with each other (Renau et al., 2019) and exhibit strong sensitivity to random sampling strategies (Renau et al., 2020). This necessitates an additional step of feature selection for the given MetaBBO scenario, requiring a certain level of expert knowledge. 2) The landscape features given by ELA are more like an "offline" portray of the target optimization problem. This is somewhat incompatible with the dynamic nature of MetaBBO paradigm, which requires both the "offline" problem properties and the "online" optimization progress. 3) Computing some of these features, particularly those related to convexity and local optimum properties, consume additional function evaluations (FEs). The consumption of the computational resource, initially reserved for the low-level optimization, ultimately degrades the final performance of the MetaBBO algorithms.

To address the above limitations, we propose Neural Exploratory Landscape Analysis (NeurELA), a novel, learnable framework designed to dynamically profile optimization status for the MetaBBO algorithms in an end-to-end manner. Our NeurELA encounters certain challenges: on the one hand, we need to carefully design a universal neural network-based landscape analyser that is compatible with a variety of MetaBBO algorithms. On the other hand, it is crucial to enable effective learning within this analyser to ensure it robustly generalizes not only to unseen optimization problems but also to previously unencountered MetaBBO algorithms. We address the challenges as follows.

1) Our research begins by designing a comprehensive operating space, $\Omega$, within which our landscape analyser $\Lambda$ functions. This space encompasses a collection of MetaBBO tasks, each designated as $\mathbb{T} := \{\mathbb{A}, \mathbb{D}\}$. Here, $\mathbb{A}$ specifies a particular MetaBBO algorithm, and $\mathbb{D}$ represents the associated set of optimization problems to meta-train and test the algorithm. The primary objective is to optimize $\Lambda$ such that it enhances the performance of various $\mathbb{A}$ across their respective problem sets in $\Omega$. Our introduction of $\Lambda$ within the comprehensive operating space $\Omega$ marks a significant advancement in MetaBBO methodologies. Unlike traditional approaches that treat each optimization algorithm and problem in isolation, $\Lambda$ leverages a holistic view, enhancing performance across a diverse set of tasks.

2) We instantiate the landscape analyser $\Lambda$ through a neural network model parameterized by $\theta$. Specifically, $\Lambda_\theta$ is implemented using a two-stage attention-based neural network. Our design enables flexible integration with most existing MetaBBO algorithms as an effective replacement for their traditional hand-crafted landscape analysis mechanisms. Given the non-differential nature of our objective in BBO, we adopt an Evolution Strategy (Hansen & Ostermeier, 2001b; Li et al., 2018) to

train $\Lambda_\theta$. This training is carried out within a multitask neuroevolution (Miikkulainen et al., 2024) paradigm, aiming to optimize $\theta$ to maximize the expected performance across MetaBBO tasks.

3) We propose using the trained $\Lambda_\theta$ in two distinct ways. First, having been trained on a diverse range of tasks, $\Lambda_\theta$ is capable of being directly applied to new, unseen MetaBBO algorithms, achieving zero-shot generalization. Second, $\Lambda_\theta$ also supports fine-tuning in conjunction with the meta-training of new MetaBBO algorithms, allowing it to adapt its performance for specific optimization challenges.

Through extensive experiments, we validate the effectiveness of NeurELA, as illustrated on the right side of Figure 1. These results demonstrate that it is possible to pre-train a universal neural landscape analyser $\Lambda_\theta$ over a group of MetaBBO tasks and then generalize it to unseen ones. Our work pioneers the integration of automatic feature extraction in MetaBBO, transforming these methods into fully end-to-end systems. This advancement significantly streamlines optimization processes, enhancing efficiency and adaptability across related fields, and reducing the need for manual expertise.

## 2 RELATED WORKS

**Meta-Black-Box Optimization.** Existing MetaBBO paradigms can be classified into following categories with respect to role they play at the low-level optimization process: 1) *Hyper-parameters configurator*: this line of works meta-trained neural network-based policy to dynamically configure the hyper-parameters of the backend BBO optimizer, e.g., the scale factor ($F$) and crossover rate ($Cr$) in Differential Evolution (DE) algorithm (Sun et al., 2021; Tan et al., 2022). Initial researches (e.g., RLPSO (Wu & Wang, 2022)) employed simple MLP structures to parameterize the meta-level policy, which was enhanced by the latest works such as GLEET (Ma et al., 2024), which facilitated Transformer (Vaswani et al., 2017) architecture to boost the sequential decision making along the low-level optimization. 2) *Update rule selector*: works in this line flexibly select desired update rules for the low-level optimization process. Several initial works learns to select one mutation operator from a pre-defined DE mutation operator pool to attain diverse optimization behaviours (Sharma et al., 2019; Tan & Li, 2021; Tan et al., 2022). Then the research scope extends to the refinement of the operators pool for optimizing more complex problems (Lian et al., 2024), and dynamically selecting the entire algorithm(Guo et al., 2024a). 3) *Complete optimizer generator*: works in this line aims to meta-learning the neural network-based policy as a complete optimizer. Lange et al. (Lange et al., 2023; 2024) proposed meta-training several attention blocks as the complete updating process of CMAES (Hansen et al., 2003) , which was followed by Song et al. (Song et al., 2024), proposing meta-training Transformer-based language model as an entire BBO method. Such end-to-end deep learning systems encountered interpretability issue. To this end, SYMBOL (Chen et al., 2024) proposed generating symbolic update rules for the low-level optimization process, promoting the interpretable and automatic discovery of black-box method. GLHF (Li et al., 2024) carefully design a Transformer-based neural network that imitates the evolutionary operators in DE algorithm. As for the designs of optimization status extraction, a great majority of existing MetaBBO algorithms borrow the idea from well-known landscape analysis researches (Wright et al., 1932; Mersmann et al., 2011). To make those landscape features compatible with the MetaBBO paradigm, existing MetaBBO algorithms more or less facilitated feature selection and feature aggregation procedure. Take a recent work GLEET (Ma et al., 2024) as an example, it aggregates the representative ELA features (Mersmann et al., 2011) into a 9-dimensional high-level profiling features, which helps reduce the training complexity and enhance the overall optimization performance. However, we have to note that such hand-crafted feature processing requires deep expertise in optimization.

**Landscape Analysis.** Landscape Analysis originates from the emergence of Automated Algorithm Selection (AAS) (Rice, 1976; Kerschke et al., 2019), which selects an optimal algorithm out of a collection of algorithms for any given optimization problem. Generally, Landscape analysis is formulated as a function: $\Lambda : (X, y) \to s$, that maps the sampled candidates $X$ and their corresponding objective values $y$ to a feature vector $s$, each dimension of which indicates some problem-specific properties of the target optimization problem. The feature vector $s$ can be subsequently fed into machine learning algorithms (e.g., Support Vector Machines (Cortes & Vapnik, 1995) and Random Forests (Breiman, 2001)) to address some specific algorithmic tasks (Kerschke & Trautmann, 2019a). Over the years, a wide range of Landscape Analysis feature groups have been developed, and structurally summarized in (Wright et al., 1932; Pitzer & Affenzeller, 2012; Malan & Engelbrecht, 2013; Kerschke & Trautmann, 2019b), of which the mostly utilized ones are: 1) *Fitness Distance*

*Correlation* (Jones et al., 1995) includes six features profiling the distances of the sampled candidates both in decision space and the objective space, as well as the correlation in-between the two spaces. 2) *Dispersion* (Lunacek & Whitley, 2006) features describe the degree of dispersity for the best quantiles of the sampled candidates. 3) *Classical ELA* (Mersmann et al., 2011) proposed using six groups of low-level metrics such as local search, skewness of objective space and approximated curvature in (second) first-order as a comprehensive abstract of basic landscape properties (e.g., global structure, multimodality, separability). Generally, classic ELA requires adequate samples and enormous computational efforts to conclude all of its fifty low-level feature values. 4) *Information Content* (Muñoz et al., 2014) comprises five features about the smoothness, ruggedness and neutrality of an optimization problem. It first applies random walk sampling (e.g., Metropolis Random Walk (Hastings, 1970)) for each of the sampled candidates, then calculates its features by statistic metrics. 5) *Nearest Better Clustering* (Kerschke et al., 2015) is a collection of features that describe the distribution of the nearest better neighbors and the nearest neighbors of the sampled candidates.

While these hand-crafted landscape analysis features provide a systematic understanding of optimization problems, they inevitably hold high computing complexity hence consume additional computational resources. Besides, the intricate correlations in-between these feature groups (Renau et al., 2019) and their sensitivities to the random sampling strategy (Renau et al., 2020) necessitate the expert-level feature selection and feature aggregation procedures. To address these issues, several early-stage studies have recently proposed learning neural networks (e.g., convolutional neural network (Prager et al., 2021a) and point cloud transformer (Seiler et al., 2022a)) as the feature mapping function. Prager et al. (2021b) first proposed using Deep-CNN to extract landscape features from fitness map of moderate initial sample points. The neural network is trained to correctly predict algorithm configuration of modular CMA-ES algorithms. However, solely feeding the fitness map might introduce information loss. To address such issue, Seiler et al. (2022b) propose a novel point cloud transformer (PCT) based approach, which embeds sample points information and the fitness map together as the input. This work generally achieves superior landscape feature prediction accuracy. Another technical bottleneck of (Prager et al., 2021b) is the supervised training, which might leads the learned landscape representation overfit to a certain AAS/AAC task. To refine this issue, van Stein et al. (2023) propose self-supervised learning through AutoEncoder, which adopted fitness map as the target to be reconstructed. More recently, Seiler et al. (2024) propose DeepELA to adrress both the infomation loss and the supervised trainng issues. It introduces MHA and trains it by a contrastive learning loss (InfoNCE) to ensure aligned representation across different sampling strategies. In this paper, our NeurELA presents several novel contributions compared with the above learning-based ELA frameworks: a) NeurELA is designed for dynamic MetaBBO tasks, enabling dynamic algorithm configuration and selection, whereas previous studies focus on static profiling of a given problem. b) NeurELA introduces a scalable embedding strategy that removes dimensional constraints in DeepELA and employs a two-stage attention mechanism for extracting features across sample points and problem dimensions, making it more generic and versatile for various MetaBBO tasks. For more detailed differences, please refer to Appendix B.2.

## 3 NEURAL EXPLORATORY LANDSCAPE ANALYSIS

### 3.1 FRAMEWORK

We introduce a multi-task operating space, denoted as $\Omega$, which serves as the foundational basis for our landscape analyser, $\Lambda$. The space $\Omega$ is mathematically defined as:

$$\Omega = \{\mathbb{T}_k = (\mathbb{A}_k, \mathbb{D}_k) \mid k = 1, 2, \ldots, K\} \tag{1}$$

where each $\mathbb{T}_k$ represents a task configuration consisting of a MetaBBO algorithm $\mathbb{A}_k$ and a corresponding set of optimization problems $\mathbb{D}_k$. This structured collection of tasks is meticulously designed to optimize our proposed landscape analyser $\Lambda_\theta$ over a variety of algorithms and problems.

In accordance with the MetaBBO protocol, each algorithm $\mathbb{A}_k$ is composed of three fundamental elements: the meta-level neural policy $\Pi_\phi$, the low-level optimizer $\Gamma$, and the initial landscape feature analyser, denoted as $\Lambda_0$. As illustrated in Figure 1, the meta-training of $\mathbb{A}_k$ adheres to a bi-level optimization paradigm. Given a problem instance $f$ sampled from $\mathbb{D}_k$, at each optimization step $t$, the low-level BBO method $\Gamma$ interacts with $f$ to attain a raw observation $\{X_i^t, y_i^t\}_{i=1}^m$ of the current population of candidates solutions, denoted as $o^t$, where $m$ is the population size in population-based

Figure 2: **Left**: The architecture of the basic attention block ($Attn$). **Right**: The computation graph of the two-stage attention mechanism (Ts-Attn).

search methods Storn & Price (1997); Kennedy & Eberhart (1995); Holland (1975). Subsequently, the observation $o^t$ is further processed by the original landscape analyser, $\Lambda_0$, to extract optimization status features $s^t = \Lambda_0(o^t)$. Based on these features, the meta-level neural policy $\Pi_\phi$ dictates a design choice $c^t$ from the configuration space $\mathbb{C}$ conditioned on the current optimization status: $c^t = \Pi_\phi(s^t)$. Then, the low-level BBO method $\Gamma$ loads $c^t$ to optimize $f$ for next time step $t + 1$. The meta-performance $r^t$ is calculated by evaluating the performance gain achieved by implementing $c_t$. The ultimate goal of the meta-training phase is to maximize the overall meta-objective $F_{\text{metaBBO}}$, defined as:

$$\max_\phi F_{\text{metaBBO}} = \mathbb{E}_{f \sim \mathbb{D}_k, \Pi_\phi} \left[ \sum_{t=0}^{T} r^t \right], \tag{2}$$

where the expectation is over all problem instances from $\mathbb{D}_k$ and spans the total length $T$ of the low-level optimization horizon.

In our NeurELA, we parameterize a neural network-based landscape analyser $\Lambda_\theta$ (where $\theta$ denote the learnable parameters), which substitutes the original landscape analyser $\Lambda_0$, to produce the optimization status features as $s^t = \Lambda_\theta(o^t)$. Our aim is to develop a universal $\Lambda_\theta$ for diverse MetaBBO algorithms that perform comparably to, or better than, the original hand-crafted $\Lambda_0$ in the corresponding MetaBBO tasks. To objectively measure the performance of $\Lambda_\theta$, we define an evaluation metric $\Upsilon(\Lambda_\theta | \mathbb{T}_k)$ as the relative performance of $\Lambda_\theta$ against $\Lambda_0$ in the context of MetaBBO task $\mathbb{T}_k = (\mathbb{A}_k, \mathbb{D}_k)$. Concretely, we first meta-train two versions of the MetaBBO algorithms on $\mathbb{D}_k$: $\mathbb{A}_k(\Pi_\phi, \Gamma, \Lambda_0)$ and $\mathbb{A}'_k(\Pi_\phi, \Gamma, \Lambda_\theta)$, and then evaluate $\Upsilon(\Lambda_\theta | \mathbb{T}_k)$ as:

$$\Upsilon(\Lambda_\theta \mid \mathbb{T}_k) = \frac{1}{P \times Q} \sum_{p=1}^{P} \sum_{q=1}^{Q} Z_{p,q}, \text{ where } Z_{p,q} = -\frac{f^*_{p,q} - \mu_p}{\sigma_p} \tag{3}$$

where $P = |\mathbb{D}_{\text{test}}|$ is the number of test problems (the dataset $\mathbb{D}_k$ is divided into two distinct subsets: $\mathbb{D}_{\text{train}}$ for training and $\mathbb{D}_{\text{test}}$ for testing), $Q = 50$ represents the number of independent runs to obtain average performance, $f^*_{p,q}$ denotes the objective value achieved by testing $\mathbb{A}'_k$ on the $p$-th problem instance and $q$-th run, and $\mu_p$ and $\sigma_p$ are the mean and standard deviation of the objective values obtained by testing $\mathbb{A}_k$ on the $p$-th problem over 50 independent runs. Note that we follow the Z-score normalization design in Ma et al. (2023) to evaluate the performance gain $Z_{p,q}$.

The meta-objective of our NeurELA framework is defined as maximizing the expected relative performance of the neural network-based landscape analyser, $\Lambda_\theta$, across all MetaBBO tasks defined within the multi-task space $\Omega$. Mathematically, this is expressed as:

$$\max_\theta F_{\text{metaELA}} = \mathbb{E}_{\mathbb{T}_k \sim \Omega} \left[ \Upsilon(\Lambda_\theta \mid \mathbb{T}_k) \right] = \frac{1}{K} \sum_{k=1}^{K} \Upsilon(\Lambda_\theta \mid \mathbb{T}_k) \tag{4}$$

## 3.2 ARCHITECTURE OF $\Lambda_\theta$

Our architecture is designed for compatibility with various MetaBBO algorithms. We introduce a Population Information Embedding (PIE) module that preprocesses observations $o^t = \{X_i^t, y_i^t\}_{i=1}^{m}$, allowing $\Lambda_\theta$ to adapt to different search ranges and objective scales. Additionally, a two-stage attention-based encoder (Ts-Attn) enhances $\Lambda_\theta$'s information extraction capabilities and scalability.

---

**Algorithm 1** Pseudo code of training NeurELA

---

1: **Input**: The MetaBBO task space $\Omega$, Evolution Strategy $ES$, optimization horizon $maxGen$.
2: **Output**: Optimal neural landscape analyser $\Lambda_{\theta*}$
3: Initialize a group of $\{\Lambda_{\theta_i}\}_{i=1}^N$ by $ES.init()$;
4: **for** $generation = 1$ **to** $maxGen$ **do**
5:     **for** each $\Lambda_{\theta_i}$ **do**
6:         **for** each MetaBBO task $\mathbb{T}_k \in \Omega$ **do**
7:             Construct $\mathbb{A}_k(\Pi_\phi, \Gamma, \Lambda_0)$ and $\mathbb{A}'_k(\Pi_\phi, \Gamma, \Lambda_\theta)$;
8:             Meta-train $\mathbb{A}_k$ and $\mathbb{A}'_k$ on $\mathbb{D}_{train}$;
9:             Test meta-trained $\mathbb{A}_k$ and $\mathbb{A}'_k$ on $\mathbb{D}_{\text{test}}$;
10:         **end for**
11:         Evaluate the fitness $f_i$ of $\Lambda_{\theta_i}$ by $F_{\text{metaELA}}$ in Eq. (4);
12:     **end for**
13:     Set $\Lambda_{\theta*}$ as the one with the highest fitness so far;
14:     Update the distributional parameters in $ES$ by $ES.update(\{\Lambda_{\theta_i}\}_{i=1}^N, \{f_i\}_{i=1}^N)$;
15:     Sample a new generation of $\{\Lambda_{\theta_i}\}_{i=1}^N$ by $ES.sample()$;
16: **end for**

---

**PIE.** PIE normalizes observation $o^t$ using two min-max normalization operations: first on the candidate solutions $\{X_i^t\}_{i=1}^m$ against the search range, and second on the objective values $\{y_i^t\}_{i=1}^m$ using the extremum values at time step $t$. This ensures unified representation and generalization by scaling all values to $[0, 1]$. For a $d$-dimensional optimization problem, the normalized observations $o^t$ are then reorganized into per-dimensional tuples $\{\{(X_{i,j}^t, y_i^t)\}_{i=1}^m\}_{j=1}^d$, with a shape of $d \times m \times 2$. These tuples undergo a linear transformation via a mapping matrix $W_{\text{emb}} \in \mathbb{R}^{2 \times h}$, producing the population information encoding $E^t$, which has the shape $d \times m \times h$. Here, $h$ represents the hidden dimension used in the subsequent two-stage attention module.

**Ts-Attn.** The computation graph of the Ts-Attn is illustrated on the right side of Figure 2, with a basic component being the attention block ($Attn$) of hidden dimension $h$. As illustrated in the left side of Figure 2, the $Attn$ block follows the design of the original Transformer Vaswani et al. (2017), except that the layer normalization Ba et al. (2016) is used instead of batch normalization Ioffe & Szegedy (2015). Ts-Attn receives $E^t$ and then advances the information sharing at both cross-solution and cross-dimension levels. **1) Cross-solution attention:** We utilize an $Attn$ block to enable same dimensions shared by different candidate solutions within the population to exchange information. **2) Cross-dimension attention:** A second $Attn$ block is implemented to further promote the sharing of information across different dimensions within each candidate. To achieve this, the output from the cross-solution phase must be transposed into the shape $m \times d \times h$ and augmented with cosine/sine positional encodings to maintain the dimensional order within a candidate. After these two phases of information processing, the resulting output $s^t$ provides comprehensive optimization status features, suitable for different MetaBBO algorithms. This highly parallelizable, attention-based architecture significantly boosts the scalability of NeurELA as the number of candidates $m$ or dimensions $d$ increases. Additional technical details about the architecture of $\Lambda_\theta$ can be found in Appendix A.1.

### 3.3 TRAINING METHOD

**Neuroevolution of $\Lambda_\theta$.** The meta-objective of our NeurELA, as defined in Eq. (4), is non-differentiable, since the evaluation of the relative performance $\Upsilon(\Lambda_\theta \mid \mathbb{T}_k)$ in Eq. (3) is not directly derived from the parameter $\theta$ of $\Lambda_\theta$. We in turn employ an Evolution Strategy, denoted as $ES$, to neuroevolve $\Lambda_\theta$ towards maximizing its expected relative performance for multiple MetaBBO tasks in $\Omega$. The pseudo code of the training is presented in Algorithm 1. The $ES$ first initializes a group of $N$ landscape analysers, then iteratively searches for $\theta^*$ through $maxGen$ generations of optimization (lines 4 to 16). Note that the meta-training of the $K$ MetaBBO tasks, which involves training their neural-policies over the training problem set, is nested within the optimization loop (lines 5 to 12) and is extremely time-consuming. We hence employ **Ray** (Moritz et al., 2018), an open-source framework for parallel processing in machine learning applications, to distribute the meta-training pipelines to multiple CPUs. This parallelization significantly enhances the overall efficiency of training NeurELA.

**Zero-shot Generalization.** The pre-trained $\Lambda_{\theta^*}$ demonstrates robust performance boosts across $K$ representative MetaBBO tasks, enabling seamless integration into unseen MetaBBO tasks. Specifically, $\Lambda_{\theta^*}$ can replace the original $\Lambda_0$ in an unseen MetaBBO algorithm, $\mathbb{A}_{\text{unseen}}$. This enables $\mathbb{A}_{\text{unseen}}$ to concentrate on refining its meta-level neural policy $\Pi_\phi$ during training on the designated problem set, utilizing the capabilities of $\Lambda_{\theta^*}$ without requiring any adaptation.

**Fine-tuning Adaptation.** If $\Lambda_{\theta^*}$ does not perform well in zero-shot generalization, we provide a flexible alternative: fine-tuning it for unseen MetaBBO tasks. In this case, $\Lambda_{\theta^*}$ is integrated into $\mathbb{A}_{\text{unseen}}$ and co-trained with its $\Pi_\phi$ during its meta-training process. This method adjusts $\Lambda_{\theta^*}$ to better align with the specific characteristics of $\mathbb{T}_{\text{unseen}}$.

## 4 EXPERIMENTAL RESULTS

In this section, we discuss the following research questions: **RQ1**: Can NeurELA generalize the pre-trained landscape analyser $\Lambda_{\theta^*}$ to unseen MetaBBO tasks? **RQ2**: If the zero-shot performance is not as expected, is it possible to efficiently fine-tune NeurELA for the underperforming MetaBBO task? **RQ3**: What does NeurELA have learned? **RQ4**: How efficient is the pre-trained landscape analyser in computing landscape features? **RQ5**: How to choose the Evolutionary Strategy for training our NeurELA? **RQ6**: What is the relationship between the model complexity and the generalization performance? Below, we first introduce the experimental settings and then address RQ1 $\sim$ RQ6.

**Training Setup.** The operating space $\Omega$ comprises $K = 3$ MetaBBO tasks. In specific, we first select three representative MetaBBO algorithms LDE Sun et al. (2021), RLEPSO Yin et al. (2021)and RL-DAS Guo et al. (2024a) from existing MetaBBO literature. Then we set the associated problem set $\mathbb{D}_k$ for these MetaBBO algorithms as the BBOB testsuites in COCO Hansen et al. (2021), which includes a variety of optimization problems with diverse landscape properties. We have to note that the selected algorithms cover diverse MetaBBO scenarios such as auto-configuration for control parameters and auto-selection for evolutionary operators, hence ensuring the generalization of our NeurELA. We follow the original settings of these MetaBBO algorithms during their meta-training, as suugested in their papers. For the settings about the neural network, we set the hidden dimension $h = 16$ for the neural network modules in the landscape analyser $\Lambda_\theta$. With a single-head attention in $Attn$, $\Lambda_\theta$ possess a total number of 3296 learnable parameters. We adopt Fast CMA-ES Li et al. (2018) as $ES$ for its searching efficiency and robust optimization performance. During the training, we employ a population of $N = 10$ $\Lambda_\theta$s within a generation and optimize the population for $maxGen = 50$ generations. For each generation, we parallel the $N \times K = 30$ meta-training pipelines across 30 independent CPU cores using **Ray** Moritz et al. (2018). For each low-level optimization process, we set the maximum function evaluations as 20000. The experiments are run on a computation node of a Slurm CPU cluster, with two Intel Sapphire Rapids 6458Q CPUs and 256 GB memories. Due to the limitation of the space, we present more technical details such as the train-test split $\{\mathbb{D}_{\text{train}}, \mathbb{D}_{\text{test}}\}$, the control parameters of Fast CMAES and the use of open-sourced software in Appendix A.2 $\sim$ A.5.

**Baselines.** We consider three baselines for comparison: **1) Original**, which denotes the original hand-crafted landscape analyser functions ($\Lambda_0$) in the MetaBBO algorithms. **2) ELA**, which denotes the traditional ELA feature groups summarized in Section 2 and defined in (Mersmann et al., 2011), which we implement by the open-sourced Pflacco package (Kerschke & Trautmann, 2019b). **3) DeepELA** (Seiler et al., 2024), which is a recent work which trains an attention-based neural network as an alternative of traditional ELA features through self-supervised contrastive learning.

### 4.1 ZERO-SHOT PERFORMANCE (RQ1)

Recall that a MetaBBO task is composed of an MetaBBO algorithm and an associated optimization problem set (i.e., $\mathbb{T}_k = (\mathbb{A}_k, \mathbb{D}_k)$). We hence generalize the pre-trained landscape analyser $\Lambda_{\theta^*}$ towards three zero-shot cases: 1) five unseen MetaBBO algorithms DEDDQN Sharma et al. (2019), RLPSO Wu & Wang (2022), GLEET Ma et al. (2024), MelBA (Chaybouti et al., 2022) and GLHF (Li et al., 2024); 2) two unseen problem set BBOB-Noisy Hansen et al. (2021) and Protein-Docking benchmark Hwang et al. (2010); 3) both the unseen MetaBBO algorithms and problem sets. Those tested MetaBBO tasks cover a wide range of meta-learned black-box optimizers. Besides, the problem sets include challenging realistic application, e.g., Protein-Docking. We evaluate the performance of

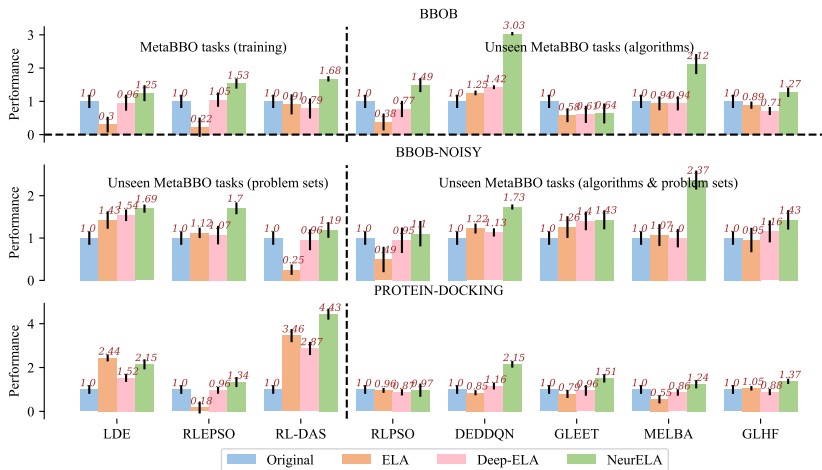

Figure 3: Zero-shot performance of NeurELA on unseen MetaBBO algorithms and problem sets.

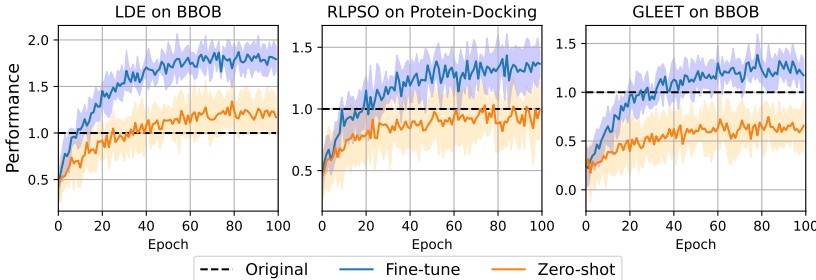

Figure 4: Performance gain curves of fine-tuning NeurELA for a specific MetaBBO task.

the ELA, DeepELA and our NeurELA by calculating the relative optimization performance (Eq. (3)) against the Original baseline in the given MetaBBO tasks. Since the performance of Original baseline would be 0 within this setting, we report the exponential of the calculated relative performances in Figure 3, where the bars denote the average relative performance, the error bars denote the 95% bootstrapped confidence interval, for 10 runs of meta-training. For BBOB-Noisy set, we set the maximum function evaluations as 20000, while for Protein-Docking set, we set it to 1000. The three zero-shot cases are split by the dashed lines. The results in Figure 3 show that: 1) Over all of the three zero-shot cases, NeurELA generally achieves superior performance against original feature extraction designs (Original) in these MetaBBO algorithms, as well as the traditional landscape analysis features (ELA) and DeepELA. 2) The results of integrating ELA into the evaluated MetaBBO tasks are quite noisy, which might be due to the intricate correlations in-between the feature groups. Such strong correlations might challenge the information processing ability of the neural policies $\Pi_\phi$ in some MetaBBO algorithms. 3) We observe that our NeurELA significantly boosts the performance of Original baseline on unseen MetaBBO algorithms & problem sets case, which not only validates the effectiveness of our design, but also indicates that the specific feature extraction designs (Origin) in existing MetaBBO algorithms might overfits to the problem set it is meta-trained on. 4) Compared with traditional ELA, DeepELA show similar performance, which is in accordance with its the training objective to serve as automated alternative of traditional ELA features. 5) Our NeurELA consistently outperforms the traditional ELA features and DeepELA on the BBOB-Noisy testsuites and Protein-Docking benchmark, which comprises a wide range of optimization problems with either different noise models and levels (e.g., Gaussian noise and Cauchy noise Hansen et al. (2009)) or intricate realistic problem features. It indicates that NeurELA shows robust generalization ability to unseen MetaBBO tasks, which involve unseen MetaBBO algorithm & realistic problems.

## 4.2 FINE-TUNING PERFORMANCE (RQ2)

We select three representative MetaBBO task from the zero-shot performance analysis (Figure 3) to validate fine-tuning adaption efficiency of our NeurELA. The selected MetaBBO tasks includes: 1)

LDE on BBOB problem set, where NeurELA surpasses the Original baseline in LDE. 2) RLPSO on Protein-Docking problem set, where NeurELA equally performs against the Original baseline in RLPSO. 3) GLEET on BBOB problem set, where NeurELA does not achieves the expected zero-shot performance. Concretely, we integrate the pre-trained $\Lambda_{\theta*}$ into the corresponding MetaBBO algorithm, then fine-tune it in conjunction with the meta-training over the given problem set. We depict the performance gain curves during the meta-training in Figure 4. We additionally plot the performance gain curves of the meta-training during the zero-shot generalization for comparison. The results showcase the flexible adaption potential of our NeurELA: 1) fine-tuning NeurELA on specific MetaBBO tasks where our NeurELA outperforms, equally performs or underperforms Original baseline would consistently introduce significant performance boosts. 2) the fine-tuning strategy is efficient in adapting NeurELA for new MetaBBO dynamics and problem structures, which only requires less than 20% meta-training epochs to achieve the similar zero-shot performance.

## 4.3 WHAT HAS NEURELA LEARNED? (RQ3)

In this section, we presents the advantage of the landscape features learned by NeurELA against traditional ELA features under the MetaBBO settings. Concretely, we take LDE (Sun et al., 2021) as a showcase. In LDE, the meta-level policy (a LSTM-based RL agent) outputs the mutation strength factor $F^t \in (0,1)$ for the low-level DE per optimization step $t$. $F^t$ indicates the exploration-exploitation tradeoff during the optimization (larger value indicates exploration, vice versa). We first record the per step populations during the optimization process of LDE (10 runs on 10-D Rastrigin in CoCo-BBOB testsuites). We then simply classify the recorded populations as exploration population ($F^t > 0.5$) and exploitation population ($F^t \leq 0.5$). We now illustrate the low-rank visualization of the ELA feature and our NeurELA feature for the recorded populations in Figure 5. The data points denoted the 2-dimensional features obtained by first feeding the populations into ELA/NeurELA and then using PCA (Shlens, 2014) transformation on the output landscape features. Red points and blue points represent the features for the exploration and exploitation population respectively. The results validate that NeurELA is more sensitive than traditional ELA features on low-level optimization dynamics. Tra-

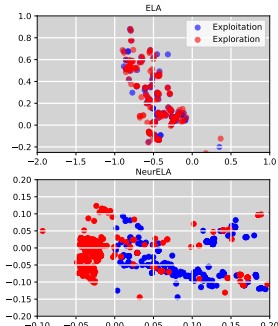

Figure 5: Low-level optimization features obtained by traditional ELA and our NeurELA, through PCA dimension reduction.

ditional ELA is mainly used for describing a problem, it might falls short in providing informative low-level optimization status for MetaBBO methods. This explains the consistent superior performance of our NeurELA (see Figure 3). Notably, our NeurELA can be meta-trained on MetaBBO tasks to provide sensitive and accurate low-level optimization status for the meta-level policy. We also provide a further interpretation analysis which reveals the correlation of features learned by NeurELA and traditional ELA features, refer to Appendix B.3 for detailed discussion.

## 4.4 IN-DEPTH STUDY (RQ4 ~ RQ6)

**Computational Efficiency Comparison (RQ4).** We compare the computational efficiency of our NeurELA with Original and ELA as either the amount of candidate samples $m$ or the dimension of the optimization problem $d$ scales. We report the average wall-time required by each baseline and our NeurELA to produce the landscape features (for 1000 runs). The results show that: 1) Our NeurELA achieves comparative computational efficiency with the specific hand-crafted designs in existing MetaBBO algorithms, while introducing significant performance boosts; 2) When the sample size $m$ increases, our NeurELA becomes more time-efficient than the original feature extraction designs in existing MetaBBO algorithms. This is primarily owning to the two-stage attention architecture in our NeurELA, which can be efficiently computed on CPUs in parallel. 3) The traditional ELA features encounters the curse of dimensionality, which makes itself impractical for MetaBBO algorithms to solve high-dimensional optimization problems.

**Evolution Strategy Selection (RQ5).** Within this paper, we employ a neuroevolution process to optimize the parameters of the landscape analyzer $\Lambda_\theta$, which presents a large-scale optimization challenge as $\Lambda_\theta$ comprises 3296 learnable parameters. To identify an effective $ES$, we implement several candidate evolution strategy variants: Fast CMAES Li et al. (2018), SEP-CMAES Ros &

Table 1: The average wall time (in seconds) for computing features.

|  | $m = 100$ | | |
|---|---|---|---|
|  | $d = 10$ | $d = 100$ | $d = 1000$ |
| Original | **4.99E-03** | **5.61E-03** | **6.38E-03** |
| ELA | 1.49E-01 | 1.48E+01 | 3.00E+02 |
| NeurELA | 5.42E-03 | 6.24E-03 | 7.11E-03 |
|  | $d = 10$ | | |
|  | $m = 100$ | $m = 500$ | $m = 1000$ |
| Original | **4.99E-03** | 2.53E-02 | 5.16E-02 |
| ELA | 1.49E-01 | 1.69E-01 | 2.02E-01 |
| NeurELA | 5.42E-03 | **7.97E-03** | **9.38E-03** |

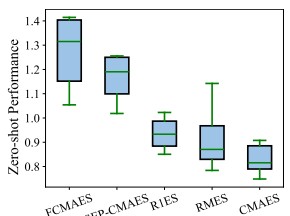

Figure 6: Zero-shot performance when using different evolution strategies.

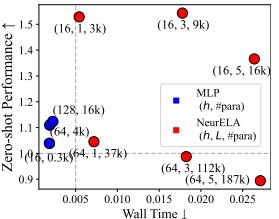

Figure 7: The influence of model complexity on the zero-shot performance.

Hansen (2008), R1ES, RMES Li & Zhang (2017), and the original CMAES Hansen & Ostermeier (2001b), and report the overall zero-shot performance (unseen algorithms & unseen problem sets) achieved by training our NeurELA with these candidate strategies for 10 runs in Figure 6. The results reveal that Fast CMAES dominates the other candidates, and is hence selected as the neural network optimizer $ES$ for our NeurELA. The convergence of these ES baselines are shown in Appendix B.1.

**Model Complexity (RQ6).** We discuss the relationship between the model complexity and the zero-shot performance (unseen MetaBBO algorithm & problem sets) of our NeurELA. Concretely, We pre-train NeurELA under 6 different model complexities, with various hidden dimensions, i.e., $h = (16, 64)$, and the number of the Ts-Attn module, i.e., $l = (1, 3, 5)$. We additionally pre-train three MLP baselines, which substitute the Ts-Attn module in NeurELA with a linear feed-forward layer, which holds a shape of $h \times h, h = (16, 64, 128)$. We report both the zero-shot performance ($y$-axis) and the computational efficiency ($x$-axis, presented as the consumed wall time for computing the landscape features) in Figure 7, where the dashed lines denotes the performance and wall-time of the Original baseline. $\#para$ denotes the number of the learnable parameters. The results show that: 1) a significant performance gap is observed between the MLP baselines and our Ts-Attn module ($h = 16, l = 1$). It validates the effectiveness of our Ts-Attn design, which enhance the feature extraction of our NeurELA by encouraging the information sharing at both the cross-solution and cross-dimension levels; 2) As the model complexity increases, the performance of the Ts-Attn module drops rapidly. It reveals that the increased number of learnable parameters challenges the optimization ability of the backend $ES$. Given the limited computational resources, it is difficult to identify the optimal parameters $\theta^*$.

**Limitations.** As a pioneer work on using deep neural network for landscape analysis in MetaBBO, our NeurELA show possibility of learning an universal landscape analyser with minimal expertise requirement and appealing optimization performance. However, a major limitation is that the training efficiency of NeurELA is not very satisfactory. As discussed in the Model Complexity part, the zero-shot performance drops rapidly when the model complexity increases. This is due to the relatively limited large-scale optimization ability of existing evolution strategies. Unfortunately, the meta-objective of NeurELA (Eq. (4)) is not differentiable hence cannot benefit from back propagation methods. We denote this limitation as an important future work to address.

## 5 CONCLUSION

In this paper, we introduce NeurELA to complement *the last mile* in making the MetaBBO paradigms entirely end-to-end: the automatic extraction of optimization status features. To this end, we propose a landscape analyser parameterized by a two-stage attention-based neural network. The analyser enables flexible replacement for the original hand-crafted landscape analysis mechanisms in existing MetaBBO algorithms. We employ a neuroevolution paradigm to optimize it by maximizing performance over a multi-task MetaBBO operating space. The experimental results showcase that our NeurELA serves two key functions: 1) it can be seamlessly integrated into unseen MetaBBO algorithms for impressive zero-shot performance, and 2) it can also be fine-tuned together with new MetaBBO algorithms to adapt its performance for specific optimization challenges. While it possesses certain limitation (e.g., low training efficiency), our work represents a significant step toward realizing a fully automatic MetaBBO algorithm—a paradigm that learns to optimize without the need for manual expertise.

ACKNOWLEDGMENTS

This work was supported in part by the National Natural Science Foundation of China No. 62276100, in part by the Guangdong Provincial Natural Science Foundation for Outstanding Youth Team Project No. 2024B1515040010, in part by the Guangdong Natural Science Funds for Distinguished Young Scholars No. 2022B1515020049, and in part by the TCL Young Scholars Program.

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

# A    TECHNICAL DETAILS

## A.1    TECHNICAL DETAILS OF $\Lambda_\theta$

Within this paper, the aim of the landscape analysis is to profile the dynamic optimization status of the current optimization process. That is, given a $d$-dimensional target optimization problem $f$, at any time step $t$, the optimization process maintains a population of $m$ candidate solutions $\{X_i^t \in \mathbb{R}^d\}_{i=1}^m$, and their corresponding objective values $\{y_i^t = f(X_i^t)\}_{i=1}^m$. We consider an end-to-end neural network structure that receives the candidates population and their corresponding objective values as input, and then outputs $h$-dimensional dynamic optimization status $s_i^t$ for each $X_i^t$. This optimization status feature aggregates the information of the optimization problem and the current candidate population, hence can be used for dynamic landscape analysis in MetaBBO algorithms. We have to note that the core challenges in designing such a neural landscape analyser locate at: 1) **generalizability**: it should be able to handle optimization problems with different searching ranges and objective value scales; 2) **scalability**: it should be capable of computing the dynamic optimization status efficiently as the amount of the sampled candidates or the dimensions of the problem scales. We address the above two challenges by designing a two-stage attention based neural network structure as the landscape analyser ($\Lambda_\theta$) in NeurELA. We now introduce the architecture of the $\Lambda_\theta$ and establish its overall computation graph step by step. For the convenience of writing, we omit superfix for time step $t$.

**Pre-processing Module.** To make NeurELA generalizable across different problems with various searching ranges and objective value scales, we apply min-max normalization over the searching space and the objective value space. Concretely, for a specific $d$-dimensional optimization problem $f$ (suppose a minimization problem), we acquire its searching range $\{[lb^j, ub^j]\}_{j=1}^d$, where $lb^j$ and $ub^j$ represent the lower bound and the upper bound at $j$-th dimension. Then we normalize each $X_i$ in the candidate population by $X_i^j = \frac{X_i^j - lb^j}{ub^j - lb^j}$, where $X_i^j$ denotes the $j$-th dimension of $X_i$. After the min-max normalization over the searching space, we min-max normalize the objective values within this time step, $y_i = \frac{y_i - y_{min}}{y_{max} - y_{min}}$, where $y_{min}$ and $y_{max}$ denotes the lowest and highest achieved objective values in this time step. We have to note that by normalizing the $X_i$ and $y_i$ within the range of $[0,1]$, we attain universal representation for different optimization problems, ensuring the generalizability of the subsequent neural network modules. The normalized $\{X_i\}_{i=1}^m$ and $\{y_i\}_{i=1}^m$ are then re-organized as a collection of meta data $\{\{(X_i^j, y_i)\}_{i=1}^m\}_{j=1}^d$ with the shape of $d \times m \times 2$. We then embed the meta data with a linear mapping $W_{emb} \in \mathbb{R}^{2 \times h}$ as the final input encoding $s$, of which the shape is $d \times m \times h$. $h$ denotes the hidden dimension of the subsequent two-stage attention module.

**Two-stage Attention Block.** We construct a two-stage attention block (Ts-Attn) to aggregate optimization status information across candidate solutions and across each dimension of the decision variables. The overall computation graph of the Ts-Attn is illustrated in the right of Figure 2 in the main body, of which a basic component is the attention block ($Attn$). As illustrated in the left of Figure 2 in the main body, the $Attn$ block mainly follows designs of the original Transformer Vaswani et al. (2017), except that the layer normalization Ba et al. (2016) is used instead of batch normalization Ioffe & Szegedy (2015). Given a group of $L$ input encoding vectors $X_{in} \in \mathbb{R}^{L \times h}$, Eq. (5) details the computation of the $Attn$ block.

$$
\begin{aligned}
g &= \text{LN}(X_{in} + \text{MHSA}(X_{in})) \\
v &= \text{FF}^{(2)}(\text{ReLU}(\text{FF}^{(1)}(g))) \\
o &= \text{LN}(g + v)
\end{aligned}
\tag{5}
$$

where MHSA, LN and FF denote the multi-head self-attention Vaswani et al. (2017) (with the hidden dimension of $h$), layer normalization Ba et al. (2016) and linear feed forward layer respectively. The output $o$ holds identical shape with the input $X_{in}$. In our Ts-Attn block, we employ an $Attn$ block $Attn_{inter}$ for the first cross-solution information sharing stage, and the other $Attn$ block $Attn_{intra}$ for the second cross-dimension information sharing stage (illustrated in the right of Figure 2 in the main body. The Ts-Attn receives the input encoding $s$ of the current candidate population, and then advances the information sharing in both cross-solution and cross-dimension level. The computation

is detailed in Eq. (6).

$$H = \text{Attn}_{inter}(S)$$
$$H = \text{Transpose}(H, d \times m \times h \to m \times d \times h) + \text{PE}$$
$$H_{out} = \text{Attn}_{intra}(H) \tag{6}$$
$$F_{indiv} = \text{MeanPooling}(H_{out}, m \times d \times h \to m \times h)$$
$$F_{pop} = \text{MeanPooling}(F_{indiv}), m \times h \to h)$$

At the first stage, we let the input encoding $s$ (attained from the pre-processing module) pass through $Attn_{inter}$. Since we group the encodings of the same dimension of all candidates in $s$, the $Attn_{inter}$ promotes the optimization information sharing across candidates in current population. From the first stage, we obtain a group of hidden features $H$ with the shape of $d \times m \times h$. At the second stage, we first transpose $H$ into the shape of $m \times d \times h$ to regroup all dimensions of a candidate together. We then add $cos/sin$ positional encoding (PE) over the transposed $H$ to inform the order of different dimensions in a candidate. We then let $H$ pass through $Attn_{intra}$ to advance the information sharing among the different dimensions within the same candidate. The output of $Attn_{intra}$ holds the shape of $m \times d \times h$. At last, we apply MeanPooling on $H_{out}$ to get the landscape feature for each candidate $F_{indiv}$ in the population, and apply a second MeanPooling on $F_{indiv}$ to get the landscape feature for the whole candidate population $F_{pop}$. We have to note that we calculate both $F_{indiv}$ and $F_{pop}$ to make our NeurELA compatible with diverse MetaBBO algorithms, which either require the landscape feature of the whole population (e.g., Wu & Wang (2022)) or require a separate landscape feature for each candidate (e.g., Sun et al. (2021)). The highly parallelizable attention-based neural-network architecture ensure the scalability of our method as the amount of the sampled candidates or the dimensions of the problem increases.

Now we summarize the end-to-end workflow of the neural landscape analyser ($\Lambda_\theta$) in our NeurELA. At any time-step $t$ within the optimization process, the pre-processing module transforms the information of the candidate population (i.e., $\{X_i^t\}_{i=1}^m$ and $\{y_i^t\}_{i=1}^m$) into the input encoding $s$. Then the Ts-Attn module transforms $s$ into the dynamic landscape features $F_{inidiv}^t$ and $F_{pop}^t$.

## A.2 TRAIN-TEST SPLIT OF BBOB TESTSUITES

BBOB contains 24 synthetic problems owning various landscape properties Mersmann et al. (2011) (e.g. multi-modality, global structure, separability and etc.). Table 2 list all of the 24 problems according to their category. Due to the diversity of properties, how to split these problems into train-test set becomes a key issue to ensure the training performance and its generalization ability. Our fundamental principle to split is to maximize the inclusion of representative landscape properties as possible. Specifically we visualize these 24 problems under 2D setting, and then select 12 representative problems into train set. We also provide contour map of problems in train set in Figure 8 and test set in Figure 9. Moreover, to avoid possible issue Kudela (2022) coming from fixed optima which is often located in $[0, ..., 0]$ in current benchmark problems (this might facilitate model to overfit to this fixed point), we thus add random offset $O$ into each problems, that is to convert $y = f(x)$ into $y = f(x - O)$. This operation is inserted into both train set $\mathbb{D}_{\text{train}}$ and test set $\mathbb{D}_{\text{test}}$.

## A.3 TRAIN-TEST SPLIT OF BBOB-NOISY AND PROTEIN-DOCKING TESTSUITES

We summarize some key characteristic of this two testsuits as follows.

- **BBOB-Noisy**: this testsuits contains 30 noisy problems from COCO Hansen et al. (2021). They are obtained by further inserting noise with different models and levels into problems in BBOB testsuits. BBOB-Noisy is characterized by its noisy nature and often used to examine robustness of certain optimizers.

- **Protein-Docking**: this testsuits contains 280 instances of different protein-protein complexes Hwang et al. (2010). These problems are characterized by rugged objective landscapes and are computationally expensive to evaluate.

We follow train-test split for these two testsuites defined in MetaBox Ma et al. (2023). Under easy mode in MetaBox, 75% of instances are allocated into training and the remaining 25% are used in

Table 2: Overview of the BBOB testsuits.

| | No. | Functions |
|---|---|---|
| | 1 | Sphere Function |
| | 2 | Ellipsoidal Function |
| Separable functions | 3 | Rastrigin Function |
| | 4 | Buche-Rastrigin Function |
| | 5 | Linear Slope |
| Functions with low or moderate conditioning | 6 | Attractive Sector Function |
| | 7 | Step Ellipsoidal Function |
| | 8 | Rosenbrock Function, original |
| | 9 | Rosenbrock Function, rotated |
| Functions with high conditioning and unimodal | 10 | Ellipsoidal Function |
| | 11 | Discus Function |
| | 12 | Bent Cigar Function |
| | 13 | Sharp Ridge Function |
| | 14 | Different Powers Function |
| Multi-modal functions with adequate global structure | 15 | Rastrigin Function (non-separable counterpart of F3) |
| | 16 | Weierstrass Function |
| | 17 | Schaffers F7 Function |
| | 18 | Schaffers F7 Function, moderately ill-conditioned |
| | 19 | Composite Griewank-Rosenbrock Function F8F2 |
| Multi-modal functions with weak global structure | 20 | Schwefel Function |
| | 21 | Gallagher's Gaussian 101-me Peaks Function |
| | 22 | Gallagher's Gaussian 21-hi Peaks Function |
| | 23 | Katsuura Function |
| | 24 | Lunacek bi-Rastrigin Function |
| Default search range: $[-5, 5]^D$ | | |

testing. Training or further fine-tuning on these two testsuites in our experiments are executed in the train set, and then validate performance of corresponding MetaBBO algorithms in the test set $\mathbb{D}_{\text{test}}$.

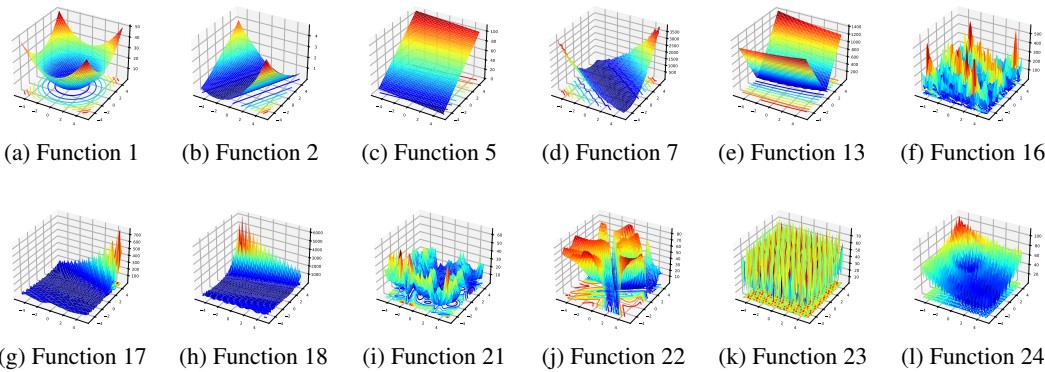

(a) Function 1  (b) Function 2  (c) Function 5  (d) Function 7  (e) Function 13  (f) Function 16

(g) Function 17  (h) Function 18  (i) Function 21  (j) Function 22  (k) Function 23  (l) Function 24

Figure 8: Fitness landscapes of functions in BBOB **train** set when dimension is set to 2.

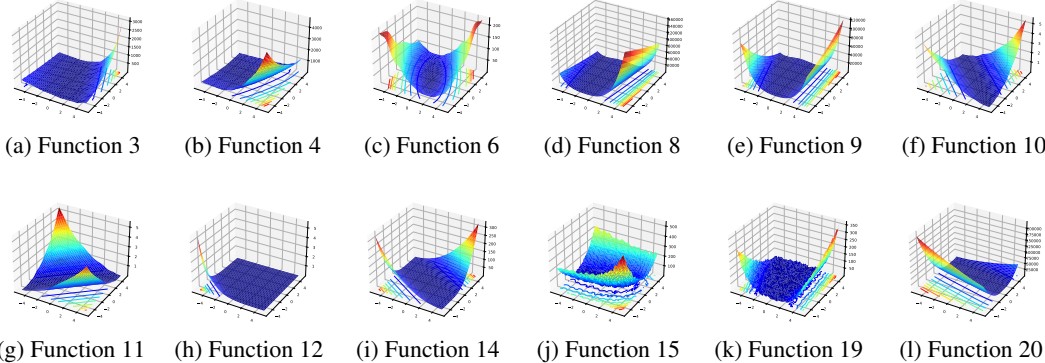

(a) Function 3  (b) Function 4  (c) Function 6  (d) Function 8  (e) Function 9  (f) Function 10

(g) Function 11  (h) Function 12  (i) Function 14  (j) Function 15  (k) Function 19  (l) Function 20

Figure 9: Fitness landscapes of functions in BBOB **test** set when dimension is set to 2.

### A.4 LICENSE OF USED OPEN-SOURCED ASSETS

Our codebase can be accessed at `https://anonymous.4open.science/r/Neur-ELA-303C`. In Table 3 we listed several open-sourced assets used in our work and their corresponding licenses.

Table 3: Used open-sourced tools and their licenses.

| Used scenario | Asset | License |
|---|---|---|
| Top-level optimizer | PyPop7 Duan et al. (2022) | GPL-3.0 license |
| MetaBBO algorithms implementation Low-level train-test workflow | MetaBox Ma et al. (2023) | BSD-3-Clause license |
| Parallel processing | Ray Moritz et al. (2018) | Apache-2.0 license |
| ELA feature calculation | pflacco Kerschke & Trautmann (2019b) | MIT license |

### A.5 CONTROL-PARAMETERS OF $ES$

**Fast CMAES** We grid-search three key hyper-parameters in Fast CMAES, including the mean value $\mu$ and sigma value $\sigma$ of the initial Gaussian distribution used for sampling, learning rate of evolution path update $c$. We list the grid search options in Table 4 and choose the best setting according to training performance on BBOB. Besides, for other control-parameters of the Fast CMAES, we follow the default settings listed in its original paper Li et al. (2018).

Table 4: Grid-search of control-parameters of Fast CMAES.

| Control-parameters | Grid options | Selected setting |
|---|---|---|
| Initial mean value $\mu$ | $[0^D, \mathcal{R}^D]$ | $\mathcal{R}^D$ |
| Initial sigma value $\sigma$ | $[0.1, 0.3]$ | $0.3$ |
| Learning rate of evolution path update $c$ | $[2.0/(D+5.0), 6.0/(D+5.0)]$ | $2.0/(D+5.0)$ |

Note: $D$ represents the searching dimension of Fast CMAES. More specifically, as the top-level optimizer to neural-evolve our neural landscape analyser $\Lambda_\theta$, $D$ specifies the dimension of $\Lambda_\theta$ which is 3296 under default settings in our main experiment.

**Other candidate evolution strategy variants** We follows the default settings as implementations in PyPop7 Duan et al. (2022) for other candidate top-level optimizers. We made a comparision study among SEP-CMAES Ros & Hansen (2008), R1ES, RMES Li & Zhang (2017), original CMAES Hansen & Ostermeier (2001b) and Fast CMAES under their default settings and finally select Fast CMAES as the default top-level optimizer of this work.

## B ADDITIONAL DISCUSSION

### B.1 TRAINING CONVERGENCE

In NeurELA, the meta-objective as defined in Eq. (4), is non-differentiable. Hence, we train the neural network in NeurELA through neuroevolution. Such paradigm requires effective evolutionary optimizers which maintain a population of neural networks and reproduce elite offsprings iteratively according to the training objective of the neural networks. In NeurELA, we adopt Evolution Strategy (ES) since it is claimed to be more effective then other optimizers. There are many modern variants of ES method, of which we select five: Fast CMAES (Li et al., 2018), Sep-CMAES (Ros & Hansen, 2008), R1ES (Li & Zhang, 2017), RMES (Li & Zhang, 2017) and CMAES (Hansen & Ostermeier, 2001b) as candidates. We present the training curves of all five optional ES baselines under our training settings in Figure 10. The results demonstrate that the Fast CMAES we adopted for

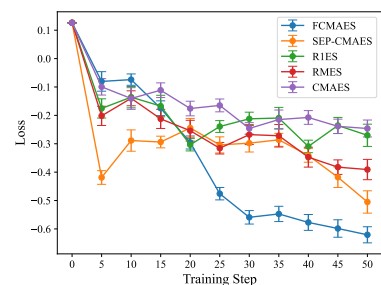

Figure 10: Training curves of different ES baselines when training NeurELA

training NeurELA converges and achieves superior training
effectiveness to other ES baselines.

## B.2 DIFFERENCE BETWEEN NEURELA AND DEEP-ELA

Although a previous work Deep-ELA (Seiler et al., 2024) also proposed using attention-based
architecture for landscape analysis, there are significant differences between our NeurELA and
Deep-ELA, which we listed as below:

**1. Target scenario.** NeurELA is explicitly designed for MetaBBO tasks, where dynamic optimization
status is critical for providing timely and accurate decision-making at the meta level. In contrast,
Deep-ELA serves as a static profiling tool for global optimization problem properties and is not
tailored for dynamic scenarios. NeurELA supports dynamic algorithm configuration, algorithm
selection, and operator selection. In contrast, Deep-ELA's features are restricted to static algorithm
selection and configuration, limiting its adaptability in dynamic MetaBBO workflows.

**2. Feature extraction workflow.** First, NeurELA addresses the limited scalability of Deep-ELA for
high dimensional problem. Concretely, the embedding in Deep-ELA is dependent on the problem
dimension and hence the authors of Deep-ELA pre-defined a maximum dimension (50 in the original
paper). To address this, NeurELA proposes a novel embedding strategy which re-organizes the sample
points and their objective values to make the last dimension of the input tensor is 2 (Section 3.2).
This embedding format has a significant advantage: the neural network of NeurELA is hence capable
of processing any dimensional problem and any number of sample points. NeurELA enhances the
information extraction through its two-stage attention-based neural network. Specifically, when
processing the embedded data, Deep-ELA leverages its self-attention layers for information sharing
across sample points only. In contrast, NeurELA incorporates a two-stage attention mechanism,
enabling the neural network to first extract comprehensive and useful features across sample points
(cross-solution attention) and then across problem dimensions (cross-dimension attention). This
design helps mitigate computational bias and improve feature representation.

**3. Training method.** The training objective and training methodology in NeurELA and Deep-
ELA are fundamentally different. Deep-ELA aims to learn a neural network that could serve as an
alternative of traditional ELA. Its training objective is to minimize the contrastive loss (InfoNCE)
between the outputs of its two prediction heads (termed as student head and teacher head) by gradient
descent, in order to achieve invariance across different landscape augmentation on the same problem
instance. In contrast, the training objective of NeurELA is to learn a neural network that could provide
dynamic landscape features for various MetaBBO tasks. Specifically, its objective is to maximize the
expected relative performance improvement when integrated into different MetaBBO methods. Since
such relative performance improvement is not differentiable, NeurELA employs neuroevolution as its
training methodology. Neuroevolution is recognized as an effective alternative to gradient descent,
offering robust global optimization capabilities.

In summary, NeurELA and Deep-ELA are two totally different works with different target operating
scenarios, algorithm design tasks, neural network designs and workflows, and training methodologies.

## B.3 FURTHER INTERPRETATION ANALYSIS

To further interpret what features have been learned by our NeurELA, we have conducted following
experimental analysis to further interpret the relationship between NeurELA features and traditional
ELA features, where we uses Pearson Correlation analysis to quantify the correlation between each
NeurELA feature and each traditional ELA feature.Below, we explain our experimental methodology
step by step:

1. We select three MetaBBO methods (LDE, RLEPSO and RL-DAS) from our training task set
and employ their pre-trained models to optimize the 24 problem instances in CoCo BBOB-10D
suite. Each MetaBBO method performed 10 independent runs per problem instance, with each run
consisting of 500 optimization steps. Now we obtain 3*24*10 = 720 optimization trajectories, each
with length 500, and the data at each step of a trajectory is the population and the corresponding
objective values $\{Xs, Ys\}$.

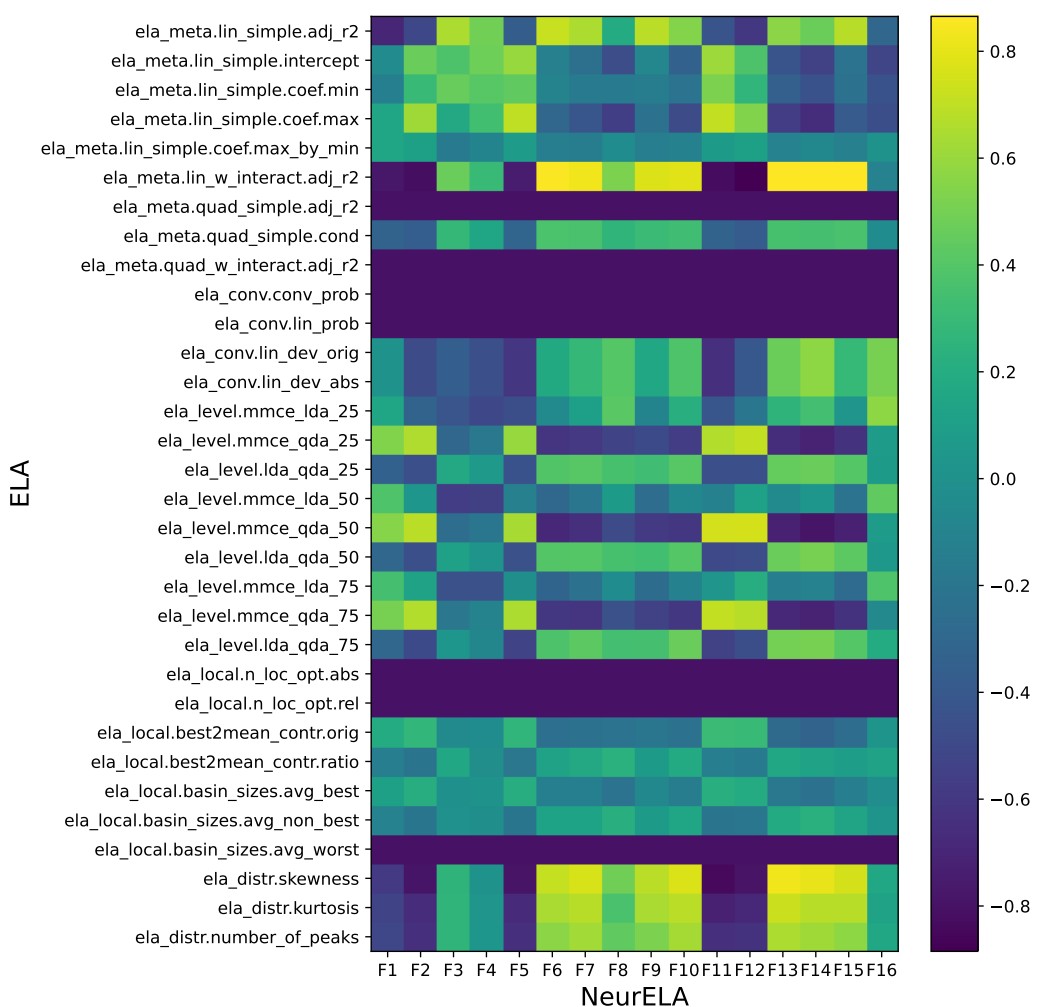

Figure 11: Correlation score of NeurELA features and traditional ELA features

2. Based on the obtained trajectories, we use the pre-trained NeurELA model (outputs 16 features) and the traditional ELA (we choose 32 ELA features from the traditional ELA including the Meta-model group, Convexity group, Level-Set group, Local landscape group and Distribution group) to calculate landscape features for each optimization step. After the computation, we obtain 720 landscape features time series for NeurELA and traditional ELA respectively.

3. For each pair of landscape features time series, we measure the relationship between the i-th feature in NeurELA features and the j-th feature in traditional ELA features by computing the Pearson Correlation Coefficient $r_{i,j}$ of the time series of these two features: $\{F_{i,1}^{NeurELA} \dots F_{i,500}^{NeurELA}\}$ and $\{F_{j,1}^{ELA} \dots F_{j,500}^{ELA}\}$.

4. We obtain the final correlation scores of each feature pair between NeurELA and traditional ELA by averaging $r_{i,j}$ of this feature pair over the 720 time series data, $i \in \{1, 2, \dots, 16\}, j \in \{1, 2, \dots 32\}$. Finally we obtain a correlation matrix with 32 rows and 16 columns. We illustrate this correlation matrix by the heatmap in Figure 11. The x-axis denote 16 NeurELA features and y-axis denote 32 ELA features, a larger value denotes the two features are closely related.

From the correlation results in that Figure, we could find some relationship patterns between our NeurELA features and the traditional ELA features: a) four NeurELA features F1, F4, F8 and F16 are novel features learned by NeurELA which show weak correlation (< 0.6) with all ELA features. b) some NeurELA features show strong correlation with one particular feature group in traditional

ELA, such as F3 with the Meta-model group. c) some NeurELA features show strong correlation with multiple feature groups in traditional ELA, such as F10 with Distribution group and Meta-model group. d) all NeurELA features show weak correlation with the Convexity group and Local landscape group, which might reveals these two group features are less useful for addressing MetaBBO tasks.

