# OpenReview forum: "Neural Exploratory Landscape Analysis for Meta-Black-Box-Optimization"
_ICLR.cc/2025/Conference — ICLR 2025 Poster_

### Official Review · Reviewer_C3y6 · 2024-10-22

**Soundness:** 3
**Presentation:** 3
**Contribution:** 3
**Rating:** 8
**Confidence:** 3

**Summary:**

The paper proposes Neural Exploratory Landscape Analysis (NeurELA), a framework to replace hand-crafted landscape analysis features in Meta Black-Box Optimization (MetaBBO) with a learned neural network approach.

The key contributions are:
- making the MetaBBO paradigm entirely end-to-end
- A two-stage attention-based neural network architecture for dynamic landscape feature extraction
- A multi-task training framework to learn from multiple MetaBBO algorithms simultaneously
- Demonstration of zero-shot generalization to unseen algorithms and problem sets
- Ability to fine-tune the pre-trained analyzer for specific tasks

**Strengths:**

- The approach is well-motivated and builds on established MetaBBO literature
- The experimental methodology is rigorous with proper baselines and ablations
- Results demonstrate clear improvements over traditional ELA features
- Comprehensive experiments across multiple MetaBBO algorithms
- Testing on both synthetic benchmarks and real-world problems (protein docking)

**Weaknesses:**

- Could better analyze when/why zero-shot generalization fails
- The authors acknowledge training efficiency issues with larger models
- Lacks theoretical justification for why the two-stage attention architecture works well

**Questions:**

- What is the minimum number of training tasks needed for good zero-shot generalization?
- Could you provide more insight into what features the network learns compared to traditional ELA?

### Nitpicking
- L110: an universal neural landscape analyser... -> a universal
- L176: (Prager et al., 2021b) first proposed... -> wrong citation type, should use \citet instead of \citep
- L318: We hence introduce Ray Moritz et al. (2018), an open-source... -> wrong citation type + you don't introduce Ray, you "employ" Ray
- L350: functiona... -> funtion
- L367: Recall that an MetaBBO task is... -> that a MetaBBO

---

> ### Author Response · Authors · 2024-11-22
> **Response to Reviewer #C3y6 (part 1/2)**
>
> We sincerely appreciate the reviewer for acknowledging our NeurELA as a well-motivated approach with rigorous, clear and comprehensive experimental analysis. We have carefully provided following point-to-point responses to clear your remaining concerns.
>
> **[W1, analyse zero-shot failure]**
>
> The zero-shot failure refers to the case where NeurELA is integrated into the MetaBBO method GLEET for the BBOB problem set (as shown on the left side of Figure 4), resulting in a performance score of 0.64, which is below expectations.  There are two main reasons which lead to this unexpected result: a) the unique and complex attention-based neural network design in GLEET. b) the relatively simpler landscapes of the tested problem instances in BBOB set. We locate the reasons by further examining the output landscape features of NeurELA for the tested problem instances in BBOB and found that the layer normalization in our design would narrow the feature value range. The narrowed feature values further go through the GLEET’s attention layers, which are further narrowed by the layer normalization there. This possibly causes the meta-level policy (GLEET’s actor network, a MLP) confused about the decision bound hence causes the unexpected performance. In contrast, when we integrate NeurELA into GLEET for BBOB-noisy set and Protein-docking set, the zero-shot perforamnce is ideal. This is because the problem instances in these sets have intricate landscapes, with significant differences that remain distinguishable even after being narrowed twice. It is also worth noting that fine-tuning NeurELA with GLEET during its meta-learning process could resolve this problem, which forces the learning of the decision bounds, as shown in the left of Figure 4. Furthermore, this fine-tuning process is very efficient since it only consume 20 training epochs to attain similar performance of the original GLEET baseline.
>
> **[W2, training efficiency]**
>
> We would like to clarify that, although a limitation of NeurELA is the training efficiency when using ES for the neuroevolution of larger models, the experimental results in our zero-shot performance (Figure 3) and in our inference efficiency (Table 1) have demonstrated that, with a relatively small model, NeurELA achieves more effective landscape feature extraction with less computational overhead than traditional ELA features and a comparable level of computational overhead to the original designs in existing MetaBBO methods.
>
> **[W3, theoretical justification of network design]**
>
> We provide an intuitive explanation why our neural network can work well here. First of all, if we look at the input and output of the traditional ELA and our NeurELA, they are same: the input is sample points and their objective values {Xs, Ys} and the output is the landscape features summarized from the input. The difference is the calculation rules. In traditional ELA, the rules are a series of human-crafted principles. In contrast, the rules in NeurELA are learned under the tailored training objective we proposed in Eq. (4). the learned neural network-based rules are inhererently more compatible with the target operating scenario: MetaBBO tasks. This is demonstrated by our experimental results (Figure 3).
>
> We also provide a straightforward explanation of the two-stage attention-based network design, which we stated at the beginning of Appendix A.1. There are three motivations: a) generalizability, the neural network should be able to handle optimization problems with different dimensions. b) scalability, the neural network should be sufficiently efficient as the number of sample points scales. c) computational completeness, the neural network should involve computation not only across sample points but also across each problem dimension. To address a), we chose attention-based network which could process any number of sample points by the attention mechanism. To address b), we chose attention-based network since attention mechanism holds highly parallelizable property. To address c), we designed the two-stage attention. The first stage is cross-solution attention, which promotes the feature computation among the sample points. The second stage is the cross-dimension attention, which further promotes the information sharing among different problem dimensions. By doing so, the neural network parameters (the attention query, key, value weights) is trained to extract useful features of the optimization status information.
>
> We hope this explanation addresses your concerns and clarifies the design and functionality of NeurELA.

---

> ### Author Response · Authors · 2024-11-22
> **Response to Reviewer #C3y6 (2/2)**
>
> **[Q1, minimal MetaBBO tasks for good zero-shot performance]**
>
> The fundamental principle of the minimal number of MetaBBO tasks for good zero-shot performance is that: it should encompass main operating scenarios of existing MetaBBO methods: dynamic algorithm configuration, dynamic operator selection, and dynamic algorithm selection. This is the reason why we choose three MetaBBO tasks as the minimal training tasks in NeurELA. As MetaBBO continues to evolve, we anticipate that this minimal requirement may increase to accommodate newly proposed operating scenarios. However, the training framework of NeurELA does not need to change.  Instead, we simply need to augment the training tasks and perform flexible re-training to adapt to the expanded requirements.
>
> **[Q2, insights of the learned features]**
>
> Following your valuable suggestion, we have added an additional experimental analysis to further explore the relationship between NeurELA features and traditional ELA features. Specifically, we use the Pearson Correlation analysis to quantify the correlation between each NeurELA feature and each traditional ELA feature. The experimental methodology, results, and corresponding discussion have been updated in the revised Appendix B.3, along with Figure 4. From the correlation results presented in the figure, we observe some notable relationship patterns between our NeurELA features and the traditional ELA features:
>
> a) Four NeurELA features (F1, F4, F8, and F16) are novel features learned by NeurELA, exhibiting weak correlation (< 0.6) with all traditional ELA features.
>
> b) Some NeurELA features strongly correlate with a specific feature group in traditional ELA, such as F3, which aligns closely with the Meta-model group.
>
> c) Some other NeurELA features strongly correlate with multiple traditional ELA feature groups, such as F10, which is highly correlated with both the Distribution and Meta-model groups.
>
> d) All NeurELA features show weak correlation with the Convexity and Local Landscape groups, suggesting these groups are less relevant for addressing MetaBBO tasks.
>
> We appreciate the reviewer for this valuable suggestion, which significantly helps improve the interpretability of NeurELA. We hope the above results and discussion could address your concern. Note: we have also added some text content into the revised paper (line 465-468, colorred in blue) to guide readers to check this interpretation analysis.
>
> **[Typos]**
>
> We sincerely apologize for our oversight during proofreading and have corrected the typos you mentioned. Additionally, we have conducted a thorough and systematic review of all text content, figures, and tables to ensure accuracy and clarity.

---

> > ### Comment · Reviewer_C3y6 · 2024-11-26
> >
> > Thank you for your detailed answer. I am satisfied with your explanations and will maintain my score.

---

> > > ### Author Response · Authors · 2024-11-26
> > >
> > > It is an honor for us. Thanks again for your precious time and efforts!

---

### Official Review · Reviewer_ARe5 · 2024-11-03

**Soundness:** 3
**Presentation:** 3
**Contribution:** 1
**Rating:** 3
**Confidence:** 3

**Summary:**

The paper introduces Neural Exploratory Landscape Analysis (NeurELA), a framework for Meta-Black-Box Optimization (MetaBBO) that replaces traditional, human-crafted Exploratory Landscape Analysis (ELA) features with a fully neural network-based approach. NeurELA employs a two-stage, attention-based neural network trained via a neuroevolution to dynamically profile optimization landscapes, adapting its features in real time. The authors show that NeurELA enhances performance across various MetaBBO algorithms and generalizes effectively to unseen tasks for zero-shot generalization and fine-tuning.

**Strengths:**

1. Quality: The paper validates the proposed method in detail by answering research questions, not only on the method's performance but also its adaptability, computational efficiency, and generalization capacity.

2. Clarity: The paper is well-structured, with clear explanations of NeurELA’s architecture, training, and integration within MetaBBO tasks.

**Weaknesses:**

1. Originality: The proposed work is very similar to Seiler et al., 2024 (Deep-ELA), which also uses multi-head attention as the main component in the architecture. The only difference seems to be that Deep-ELA uses kNN embedding, while the proposed method uses a linear transformation to encode the population information, which is widely used in LLMs to generate embedding from tokens.

2. Limited comparisons in experiments: The proposed work does not compare to any recent methods, e.g., Deep-ELA.

3. Limited tasks: Although NeurELA is tested across a variety of MetaBBO algorithms and optimization problems, the experiments lack a detailed analysis of its performance in higher-dimensional optimization scenarios, where many MetaBBO algorithms struggle.

4. Interpretability and feature analysis: Although NeurELA shows promise in dynamically adapting landscape features, there is limited discussion on the interpretability of these features in relation to traditional ELA metrics.

**Questions:**

1. What's the major difference between the proposed method and Deep-ELA? A more detailed explanation and corresponding experiments and/or ablation studies will help better support the paper's novelty and contribution.

2. Line 479: "This is primarily owning to the two-stage attention architecture in our NeurELA, which can be efficiently computed on GPUs in parallel." Was the training done on GPU? It was mentioned in line 356 that the training uses a Slurm CPU cluster.

3. The wall time is too short to really tell a difference between algorithms since programming/system processing could have a larger effect if it's in milliseconds. Please consider using a more complicate benchmark or increasing the dimensions.

---

> ### Author Response · Authors · 2024-11-22
> **Response to Reviewer #ARe5 (part 1/2)**
>
> We appreciate the reviewer for recognizing our paper being well-structured with good quality. Below, we provide point-by-point responses to address your concerns.
>
> **[W1, differences with Deep-ELA]**
>
> Thank you for raising questions regarding the distinctions between NeurELA and Deep-ELA. Although we have included **a discussion on this issue in Section 2 (lines 187–194)**, we are happy to expand and clarify the distinctions here.
>
> 1. **Target Scenario:** NeurELA is explicitly designed for MetaBBO tasks, where dynamic optimization status is critical for providing timely and accurate decision-making at the meta level. In contrast, Deep-ELA serves as a static profiling tool for global optimization problem properties and is not tailored for dynamic scenarios. NeurELA supports dynamic algorithm configuration, algorithm selection, and operator selection. In contrast, Deep-ELA’s features are restricted to static algorithm selection and configuration, limiting its adaptability in dynamic MetaBBO workflows.
> 2. **Feature Extraction Workflow**: Considering the feature extraction workflow, NeurELA distinguishes with Deep-ELA in two key aspects:
>     1. First, **NeurELA addresses the limited scalability of Deep-ELA for high dimensional problem** . Concretely, the embedding in Deep-ELA is dependent on the problem dimension and hence the authors of Deep-ELA pre-defined a maximum dimension (50 in the original paper). To address this, NeurELA proposes a novel embedding strategy which re-organizes the sample points and their objective values {Xs, Ys} to make the last dimension of the input tensor is 2 (Section 3.2, line 267-291). This embedding format has a significant advantage: the neural network of NeurELA is hence capable of processing any dimensional problem and any number of sample points.
>     2. Second, **NeurELA enhances the information extraction through its two-stage attention-based neural network**. Specifically, when processing the embedded data, Deep-ELA leverages its self-attention layers for information sharing across sample points only. In contrast,  NeurELA incorporates a two-stage attention mechanism, enabling the neural network to first extract comprehensive and useful features across sample points (cross-solution attention, lines 296–297) and then across problem dimensions (cross-dimension attention, lines 298–299). This design helps mitigate computational bias and improve feature representation.
> 3. **Training Method:** The training objective and training methodology in NeurELA and Deep-ELA are fundamentally different. Deep-ELA aims to learn a neural network that could serve as an alternative of traditional ELA. Its training objective is to minimize the contrastive loss (InfoNCE) between the outputs of its two prediction heads (termed as student head and teacher head) by gradient descent, in order to achieve invariance across different landscape augmentation on the same problem instance. In contrast, the training objective of NeurELA is to learn a neural network that could provide dynamic landscape features for various MetaBBO tasks. Specifically, its objective is to maximize the expected relative performance improvement when integrated into different MetaBBO methods. Since such relative performance improvement is not differentiable, NeurELA employs neuroevolution as its training methodology. Neuroevolution is recognized as an effective alternative to gradient descent, offering robust global optimization capabilities.
>
> In summary, NeurELA and Deep-ELA are significantly different works, with distinct target operating scenarios, algorithm design tasks, neural network designs and workflows, as well as training methodologies. We hope the above detailed explanation would clear your concern. We have added this discussion into the revised Appendix B.2 (colorred in blue). To guide readers to this discussion, we have also updated text content in Section 2, line 187-193 (colorred in blue) of the revised paper.
>
> [**W2 & Q1, comparison with Deep-ELA**]
>
> Following your suggestion, we have added Deep-ELA as a baseline in our experiments. Specifically, we utilized the open-sourced Deep-ELA model (large_50d_v1, https://github.com/mvseiler/deep_ela/tree/main/deep_ela/models), and the testing followed the same procedure used for NeurELA and other baselines. We have updated the results in Figure 3 of the revised paper. Overall, our NeurELA consistently outperforms Deep-ELA and demonstrates substantial and reliable performance improvements across the tested MetaBBO methods.

---

> ### Author Response · Authors · 2024-11-22
> **Response to Reviewer #ARe5 (part 2/2)**
>
> **[W3, high dimensional optimization scenarios]**
>
> We would like to clarify that NeurELA’s two-stage attention-based feature extractor is inherently capable of handling optimization problems of any dimensionality. As per your suggestion, we conducted additional experiments to evaluate the zero-shot performance of NeurELA on MetaBBO methods and CoCo BBOB problems with 100 and 500 dimensions. The results are presented in the following tables (Deep-ELA is excluded as it only supports up to 50 dimensions). The results demonstrate that NeurELA effectively boosts the performance of MetaBBO methods on high-dimensional problems by leveraging its generalizable and scalable two-stage attention-based feature extractor.
>
> Results on BBOB-100D
>
> |  | LDE | RLEPSO | RL-DAS | RLPSO | DEDDQN | GLEET | MELBA | GLHF |
> | --- | --- | --- | --- | --- | --- | --- | --- | --- |
> | Original | 1.0 | 1.0 | 1.0 | 1.0 | 1.0 | 1.0 | 1.0 | 1.0 |
> | ELA | 0.24 | 0.17 | 0.82 | 0.77 | 0.93 | 0.41 | 0.83 | 0.7 |
> | NeurELA | **1.42** | **1.51** | **1.76** | **1.21** | **2.86** | 0.75 | **2.06** | **1.44** |
>
> Results on BBOB-500D
>
> |  | LDE | RLEPSO | RL-DAS | RLPSO | DEDDQN | GLEET | MELBA | GLHF |
> | --- | --- | --- | --- | --- | --- | --- | --- | --- |
> | Original | 1.0 | 1.0 | 1.0 | 1.0 | 1.0 | 1.0 | 1.0 | 1.0 |
> | ELA | 0.2 | 0.19 | 0.79 | 0.62 | 0.84 | 0.51 | 0.71 | 0.55 |
> | NeurELA | **1.37** | **1.14** | **1.37** | **1.05** | **1.76** | **1.09** | **1.48** | **1.09** |
>
> **[W4, interpretability and relation with traditional ELA]**
>
> We would like to clarify that in Section 4.3, Figure 5, we have conducted an initial interpretability analysis on what NeurELA has learned compared with the traditional ELA. This analysis revealed that for a given problem, our NeurELA can provide a clear decision bound during the optimization dynamics, whereas traditional ELA cannot, since it is developed to describe static optimization problem properties rather than dynamic ones.
>
> Following your valuable suggestion, we have added an additional experimental analysis to further explore the relationship between NeurELA features and traditional ELA features. Specifically, we use the Pearson Correlation analysis to quantify the correlation between each NeurELA feature and each traditional ELA feature. The experimental methodology, results, and corresponding discussion have been updated in the revised Appendix B.3, along with Figure 4. From the correlation results presented in the figure, we observe some notable relationship patterns between our NeurELA features and the traditional ELA features:
>
> a) Four NeurELA features (F1, F4, F8, and F16) are novel features learned by NeurELA, exhibiting weak correlation (< 0.6) with all traditional ELA features.
>
> b) Some NeurELA features strongly correlate with a specific feature group in traditional ELA, such as F3, which aligns closely with the Meta-model group.
>
> c) Some other NeurELA features strongly correlate with multiple traditional ELA feature groups, such as F10, which is highly correlated with both the Distribution and Meta-model groups.
>
> d) All NeurELA features show weak correlation with the Convexity and Local Landscape groups, suggesting these groups are less relevant for addressing MetaBBO tasks.
>
> We appreciate the reviewer for this valuable suggestion, which significantly helps improve the interpretability of NeurELA. We hope the above results and discussion could address your concern. Note: we have also added some text content into the revised paper (lines 465-468, colorred in blue) to guide readers to check this interpretation analysis.
>
> **[Q2, training done on GPU?]**
>
> We apologize for the typo. The statement refers to the two-stage attention mechanism being well supported for parallelization on CPUs by pytorch. We have updated it in the revised paper.
>
> **[Q3, wall time comparison]**
>
> First, the wall time comparison focuses solely on the feature computation time for NeurELA, traditional ELA, and the original design, which is a subcomponent of the entire MetaBBO workflow, to ensure a fair and accurate comparison. Additionally, we kindly direct the reviewer’s attention to the upper part of Table 1, where we have already included results for 1000-dimensional problems—a significantly large scale. In this case, the computation time for traditional ELA is no longer in milliseconds but approximately 300 seconds. This result clearly demonstrates the superior computational efficiency of NeurELA compared to traditional ELA, even in high-dimensional settings.
>
> Finally, we sincerely thank the reviewer for all valuable comments and suggestions, which have significantly helped us improve the paper quality. We hope the above responses could clear your concerns and look forward to your positive feedback in the remaining time of the rebuttal phase.

---

> ### Author Response · Authors · 2024-11-26
> **Request for further feedback**
>
> Dear reviewer #ARe5:
>
> Since the discussion period is extended, we respectifully request you to check the experimental restults and discussion we have added following your constructive suggestions. We have given a more comprehensive discussion on the differences between our NeurELA and a recent work DeepELA to demonstrate the novelty of this work. We have shown the superiority of NeurELA to DeepELA on a) MetaBBO tasks, b) high-dimensional optimization scanarios. Furthermore, we have statistically analysed the correlation of NeurELA features and traditional ELA features to enhance interpretability. If you have any further instructions, we are open to them and would cooperate with you  to make this paper better.
>
> Best regards, the authors

---

> > ### Comment · Reviewer_ARe5 · 2024-11-27
> >
> > I appreciate the authors’ responses; however, they do not fully address my concerns. First, the differences between Deep-ELA and the proposed method appear marginal. Specifically, using common techniques such as adding positional embeddings or cross-attention and applying them to a different problem (while relying on an existing objective function) cannot be considered significantly different. Thus, the novelty of the work remains a major concern. Second, the paper lacks comparisons with other methods beyond traditional ELA, and the current results are not sufficiently convincing. Overall, I believe the paper overstates its novelty and contribution and does not meet the requirements for publication.

---

> > > ### Author Response · Authors · 2024-11-27
> > > **Response to Reviewer #ARe5**
> > >
> > > We appreciate the reviewer's timely feedback.
> > >
> > > First, we would clarify that **we have provided a detailed elaboration on differences between our NeurELA and DeepELA** in the last response ( also have added this discussion to Appendix B.2). We briefly summarize these significant novelties here:
> > >
> > > a) **We found that the reviewer overlooked one of the most significant novelty against DeepELA**: NeurELA specializes at providing dynamic landscape feature for MetaBBO tasks. In contrast, DeepELA specializes at providing static optimization problem properties for other algorithm analysis and design tasks. This is demonstrated by adding DeepELA as a new baseline and compare it with NeurELA on MetaBBO tasks (following your suggestion). The experimental results are presented in Section 4.1, Figure 3.
> > >
> > > b) **NeurELA addresses the scalability of DeepELA on high-dimensional problems** through the novel dimension-independent embedding design and the specific two-stage attention mechanism. The latter also helps fine-grained feature extraction across not only sampled points but also the dimensions within them. The experimental results of NeurELA on high-dimensional problem are provide in the last response (following your suggestion), where NeurELA outperforms the baselines in 100D and 500D problems. DeepELA can not handle such cases since its neural network structure is allowed to process maximum 50D problems.
> > >
> > > c) **NeurELA proposes a novel training objective (Section 3.1, lines 253-259, Equation (4)) and training method** **(Section 3.3, lines 308-328)** to train the proposed network strcture. Under this training objective, pre-trained NeurELA provide more useful optimization status features than baselines (demonstrated by Section 4.1, Figure 3), while comsuming less computational time (demonstrated by Section 4.4, Table 1), especially on high-dimensional and more sample points cases.
> > >
> > > Second, **we respectifully request the reviewer for specific instructions**: a) what methods should we add to compare beyond traditional ELA? b) which part of the results are not convincing? We hope for these specific instructions hence we can further address your concerns and improve our paper.
> > >
> > > At last, thanks for your time and efforts, we look forward to your further feedback.

---

### Official Review · Reviewer_PkFr · 2024-11-08

**Soundness:** 4
**Presentation:** 4
**Contribution:** 3
**Rating:** 8
**Confidence:** 3

**Summary:**

The paper introduces Neural Exploratory Landscape Analysis (NeurELA), a novel framework designed to improve Meta-Black-Box Optimization (MetaBBO) by dynamically profiling optimization landscape features through a two-stage, attention-based neural network. Unlike traditional approaches that rely on human-crafted features, NeurELA learns these features automatically in an end-to-end manner. This is aimed at overcoming the limitations of existing Exploratory Landscape Analysis (ELA) methods, such as computational overhead and reliance on expert knowledge. The authors propose a novel neural-network based landscape analyzer, and propose a two-stage attention mechanism: this architecture facilitates robust feature extraction, capable of generalizing to various MetaBBO algorithms without manual adjustments. The novel framework is tested in three MetaBBO tasks against several baselines and ablations.

**Strengths:**

- The new end-to-end pipeline is (imho) novel and interesting
- NeurELA consistently outperforms the baselines
- Many ablations are performed and thus we better understand the effectiveness of the approach and its parts
- The sharing of the code is appreciated

**Weaknesses:**

**Weaknesses before author's rebuttal. I believe that most of my comments have been (at least partly) by the new version/reply by the authors.**

- The presentation of the paper needs quite some work. Many typos are present and a few paragraphs are hard to read. Overall, the authors need to spend a bit more time in improving the presentation.
- I believe that a more detailed description of the MetaBBO tasks would greatly help the reader understand and appreciate the performance of the proposed framework.
- In Section 4.2 and Fig. 4 it is not clear to what "one epoch" corresponds to. For example, I do not understand how we can compare "ZeroShot" with "Finetune" in the same scale. If I understand correctly, the "ZeroShot" variant just uses the learned $\Lambda_{\theta}$ to select at each generation of the low-level optimizer the appropriate feature vector and configuration. On the contrary, the "Finetune" variant should run the whole meta-learning pipeline. Thus, how can we compare "epochs" of one to the other? We need more explanation here to appreciate the results.
- The computational overhead of NeurELA framework is quite big. Also compared to the original variant ($\Lambda_0$) the gains are not big even when a big number of samples ($m$) is used.

**Questions:**

See weaknesses..

---

> ### Author Response · Authors · 2024-11-22
> **Response to Reviewer #PkFr**
>
> We appreciate the reviewer for the valuable comments. We also thank you for acknowledging NeurELA as a novel and interesting work, with superior performance, convincing ablations and positive code sharing. We provide following point-to-point responses to address your remaining concerns.
>
> **[W1 & W2, presentation & writing]**
>
> Following your valuable suggestion, we have carefully checked and refined the typos in the revised paper, including the text content, figures and tables. We also agree with the reviewer that a more detailed description of the workflow in the previous section would enhance the understanding of the readers, hence, we have added some text content in the beginning of the introduction of the revised paper (line 039-046, colorred in blue) to this end.
>
> **[W3, meaning of “epoch” in Figure 4]**
>
> We would like to clarify that the “Zero-Shot” mode of NeurELA refers to integrating the pre-trained $\Lambda_\theta$ into the neural network group of a given MetaBBO method to substitute its original feature extraction mechanism. The neural network of the MetaBBO methods still requires the meta-learning process to learn a useful policy on the training problem set, while $\Lambda_\theta$ is frozen. In contrast, for the “Fine-tuning“ mode,  $\Lambda_\theta$ is activated and co-trained as part of the meta-learning process. Hence, we can plot the two performance gain curves along with the training epochs.
>
> **[W4, computational overhead]**
>
> We argue that, as shown in the top part of Table 1, the computational overhead of NeurELA is in the similar level with the original MetaBBO baselines across different problem dimensions. Moreover, as shown in the bottom part of Table 1, when the number of sample points increases, NeurELA consumes significantly less time to compute the features than the original MetaBBO baselines. We kindly invite the reviewer to examine these results. Considering the consistent performance improvements demonstrated in Figure 3, along with the comparable feature computation wall time, we believe this instead underscores the contribution of our NeurELA.

---

> > ### Comment · Reviewer_PkFr · 2024-11-25
> >
> > I am satisfied by the authors' replies and updated text/supplementary. Thus I increased my score by one point.
> >
> > My only comment is that the authors should make the meaning of "epoch" and "zero-shot mode" clearer in the text.

---

> ### Author Response · Authors · 2024-11-25
> **Thanks for Reviewer PkFr**
>
> We sincerely appreciate your positive feedback on our NeurELA! Thanks for the time and efforts you have contributed to improve our paper.

---

### Official Review · Reviewer_RDeR · 2024-11-09

**Soundness:** 3
**Presentation:** 2
**Contribution:** 3
**Rating:** 8
**Confidence:** 4

**Summary:**

This paper proposes an automatic construction method of landscape features for meta black-box optimization (MetaBBO). The proposed approach, termed neural exploratory landscape analysis (NeurELA), trains the attention-based neural network extracting landscape features to improve the MetaBBO algorithms. While existing MetaBBO methods rely on human-crafted landscape features, the proposed NeurELA can automate the design process for landscape features in MetaBBO. The experimental result demonstrates that the proposed NeurELA outperforms existing MetaBBO methods on unseen MetaBBO tasks. In addition, the authors show that fine-tuning process can lead to further performance improvement.

**Strengths:**

- The motivation for optimizing the feature extractor in MetaBBO is reasonable.
- The technical novelty of this paper is to present automating approach for extracting landscape features in MetaBBO.
- The effectiveness of the proposed NeurELA is experimentally demonstrated for several MetaBBO and BBO problems.

**Weaknesses:**

- The proposed formulation seems to be a tri-level optimization problem of training landscape feature extractor, training meta-level policy, and optimizing a target objective function. Therefore, using the proposed NeurELA increases the whole computational cost compared to existing MetaBBO methods.
- Training the landscape feature extractor is performed in a neuroevolution manner. It seems hard to scale for a large neural network as the feature extractor. In addition, it is not clear that the current setting, i.e., optimizing 3,296 parameters for 500 evaluations by the evolution strategy, is sufficient for convergence.

**Questions:**

1. I suppose that the computational cost of the proposed approach is larger than the existing MetaBBO because it requires the training of a landscape feature extractor as the outer loop for MetaBBO. What is the exact computational time/cost for the proposed method compared to existing MetaBBO methods?
1. If the baseline methods can use the same computational budgets, how is the performance gain of the proposed approach? For instance, extra budgets may be used to optimize hyperparameters or to select traditional ELA features in the baseline MetaBBO.
1. Training the landscape feature extractor in the current setting seems challenging because it should optimize the high-dimensional parameters by the evolution strategy. Is there any empirical evidence for the convergence of ES as the outer loop optimizer?
1. What kind of BBO or MetaBBO algorithms can be used with the proposed NeurELA? The authors might assume the population-based BBO algorithms or evolutionary algorithms. Is it possible to combine the NeurELA with other kinds of BBO methods?
1. What is the exact number of dimensions of the NeurELA features in the experiment? What is the impact of the dimensionality of the EeurELA features on the performance?
1. I could not find the definition of the performance metric in the experimental evaluation, e.g., in Figure 3. Could you provide a detailed definition of the performance metric in the experimental results?

---

> ### Author Response · Authors · 2024-11-22
> **Response to Reviewer #RDeR**
>
> We appreciate the reviewer for acknowledging our NeurELA as a novel landscape analysis framework with reasonable motivation and effective performance. For your remaining concerns, we provide following point-to-point responses to address them.
>
> **[W1 & Q1 & Q2, increased computational cost]**
>
> We would clarify that although certain computational cost is required to train NeurELA (through neuroevolution on multiple MetaBBO methods and downstream BBO problems), we have demonstrated in the experiments (Figure 3, zero-shot performance) that the trained NeurELA can be  seamlessly integrated into existing MetaBBO methods to provide effective dynamic landscape analysis, without further re-training. That is, NeurELA can be regarded as a feature extractor exactly the same as the traditional ELA. We also provide the inference wall time comparison in Table 1 to compare the computational cost required to obtain the landscape feature by our NeurELA and traditional ELA, where the results show that NeurELA require less processing time than traditional ELA, particularly for the high-dimensional problem, this is due to the attention-based neural network which facilitates highly paralleled computation. We believe this explanation could clear your concern in Q1 and Q2.
>
> **[W2 & Q3, training convergence]**
>
> We have included the training convergence curves of various ES baselines in the revised Appendix, as shown in Figure 3. We kindly request the reviewer to review the results in the newly added Appendix B.1 (colorred in blue), which show that under our setting, the Fast CMAES adopted for training NeurELA converges and achieves superior training effectiveness to other ES baselines.
>
> **[Q4, usage scope of NeurELA]**
>
> We would like to clarify that NeurELA mainly focuses on population-based BBO paradigm since the two-stage attention-based feature extractor we proposed is designed to process the information of a collection of sample points by first promoting the information sharing among all sample points and then promoting the relationship detection across dimensions. By doing this, we can provide accurate and dynamic landscape feature for the subsequent MetaBBO tasks. We believe NeurELA could be integrated into some human-crafted population-based BBO methods, which requires landscape feature for dynamic algorithm configuration or operator selection. This is supported by our interpretability analysis in Section 4.3, Figure 5. We can observe the clear decision bound of our NeurELA features when profiling the dynamic optimization status. This clear decision bound could serve as useful features for those human-crated population-based BBO methods.
>
> **[Q5, dimension of NeurELA features]**
>
> The dimension of NeurELA features is 16, which we have stated in Section 4, line 347-348. We have discussed the dimensionality of NeurELA features in the model complexity discussion part (Section 4.4, line 503-517), where we compare the feature dimensions 16, 64 and 128. Due to the limitations of the ES baseline in effectively searching for the optimal NeurELA neural network, increasing the feature dimension could not lead to performance improvement.
>
> **[Q6, performance metric definition]**
>
> We would like to clarify that we have provided the performance metric definition at the beginning of Section 4.1, line 373-377. It is exactly the exponential of the relative performance we have designed in Eq. (3). Under this setting, the performance of the original MetaBBO method is always 1. For our NeurELA and traditional ELA, a performance value larger than 1 indicates that substituting the original feature extraction design by NeurELA or traditional ELA could improve the optimization performance, and vice versa.

---

> > ### Comment · Reviewer_RDeR · 2024-12-03
> >
> > Thank you for the response and updated paper. The authors' response solved my concerns. Therefore, I would increase my score.

---

> > > ### Author Response · Authors · 2024-12-03
> > >
> > > We sincerely appreciate your positive feedback on our NeurELA! Thanks for the time and efforts you have contributed to improve our paper.

---

> > > ### Comment · Reviewer_RDeR · 2024-12-03
> > >
> > > This is just a comment. Perhaps the current paper title, "Neural Exploratory Landscape Analysis," might be confusing because the target application of the proposed method is not clear. To clarify the difference with the existing Deep-ELA in the title, a more explicit title might be appropriate, such as "Neural Exploratory Landscape Analysis for Meta-Black-Box Optimization."

---

> > > > ### Author Response · Authors · 2024-12-03
> > > >
> > > > We appreciate your suggestion! We agree that the suggested title might be more appropriate and would adopt it if we could change our title in the final version of the paper. Thanks for such valuable suggestion!

---

### Meta-Review · Area_Chair_Lg5i · 2024-12-20

**Metareview:**

The goal of the work is to learn a landscape featurizer for meta-blackbox optimization. The pipeline of the meta-BBO loop basically consists of:

1. A featurizer (e.g. attention-based architecture) which takes in the current trajectory of evaluations $(x,y)$ and outputs a feature.
2. This feature is then used to condition a separate blackbox optimizer (which itself needs to be neural-network based, e.g. an LSTM)
3. Since this bilevel problem is not differentiable through the featurizer, non-gradient based optimization methods (e.g. CMA-ES) must be used to train the featurizer.
4. The featurizer can be frozen and used directly over new test cases, or can also be fine-tuned if the new test situation has a large distribution shift from training (e.g. objective functions are different, or inner-loop algorithm is different)

Experiments are conducted over BBOB (deterministic and noisy) and protein design objectives, as well as when the inner-loop algorithm is changed.

Ablations show:
1. The Neural ELA features are cleanly separable for determining when the inner-loop algorithm should exploit or explore, whereas traditional ELA features are not so separable.
2. If the featurizer has too many learnable parameters, performance degrades since e.g. CMA-ES may suffer in high dimension, and that variants like FCMAES are the best for learning such parameters.

## Strengths
* After parsing the paper properly, the general logic makes sense. It's understandable why e.g. the featurizer needs to be trained with CMA-ES, and the whole setup is intuitive.
* It's interesting that these NeuralELA features can be transferrable (and worst-case just needs a bit of fine-tuning) to new algorithms as well, and not just new objective functions. This means that it's possible to construct universal features for representing the objective landscape, which can be sent to any inner-loop optimizer.

## Weaknesses
* I needed to re-read the paper multiple times to understand the nuances - The writing could be made much more clean. Currently the paper is too wordy and doesn't give enough "breathing space" between its text. Also, the notation can be reduced significantly, and I think the main method can be described much more simply in at most half a page.
* While the paper makes solid contributions, due to the way the paper is written at the moment, there aren't a lot of fundamental or profound conclusions. The paper could've been elevated, if e.g. training was done over a much larger set of algorithms and objectives, and that the featurizer's output became more universal and no longer required fine-tuning at all.

Given the Reviewer scores, for now we can definitely accept the paper for poster presentation. I wouldn't move it to spotlight however, due to the weaknesses raised above.

**Additional Comments On Reviewer Discussion:**

The reviewer scores were (3,8,8,8), leading to a clear acceptance.

The common issues raised were:
  * The paper needs to be written better and cleaner, and I agree - this can be fixed during camera-ready version, though.
  * Computational time, i.e. there is a large upfront cost of pretraining the featurizer, and possibly more if finetuning is involved. There is also the cost of using the featurizer itself at inference time, although it is quite small already.
    * The upfront cost of pretraining a meta-learned BBO system will always occur and is reasonable that it's required.
  * Since the training algorithm (e.g. CMA-ES) is zeroth order, it will naturally suffer over higher dimensions.
    * I personally disagree with this point, seeing as how e.g. ES-based algorithms have been shown to even optimize millions of parameters. I suspect that the specific CMA-ES chosen by the authors isn't optimal for high-D weight training, but they could've used other ES algorithms which are better suited.

Reviewer ARe5 (score of 3) mostly raised an issue of novelty - i.e. there seems to be similar work in this area called "DeepELA", but the authors replied with multiple comparisons to DeepELA, stating that:
  * DeepELA is for profiling optimization landscapes and not necessarily used by algorithms themselves, while NeuralELA is much more suitable for conditioning algorithms. This means all the downstream mechanics are also different (training objective, training method, etc.)
  * NeuralELA supports much higher dimensional problems (500+), while DeepELA supports at most 50.

---

> ### Public Comment · ~Zeyuan_Ma1 · 2025-02-24
> **Change of Title**
>
> Dear AC #Lg5i:
>
> Following the suggestion from the meta-review and our reviewers, we have changed title of this paper from "Neural Exploratory Landscape Analysis" to "Neural Exploratory Landscape Analysis for Meta-Black-Box-Optimization", which further clarify the scope of our paper and could help future readers more. We would like to note that this change would not harm any content consistency with the previous version. We appreciate your contribution on ICLR and particularly on our paper, thanks!
>
> Best regards,
> the authors

---

### Decision · Program_Chairs · 2025-01-22

Accept (Poster)